# A human liver cell-based system modeling a clinical prognostic liver signature for therapeutic discovery

Emilie Crouchet[1,2], Simonetta Bandiera[1,2], Naoto Fujiwara [3], Shen Li[4], Hussein El Saghire[1,2], Mirian Fernández-Vaquero[5,6], Tobias Riedl[5,6], Xiaochen Sun[3], Hadassa Hirschfield[3], Frank Jühling[1,2], Shijia Zhu [3], Natascha Roehlen[1,2], Clara Ponsolles[1,2], Laura Heydmann[1,2], Antonio Saviano[1,2,7], Tongqi Qian[3], Anu Venkatesh[3], Joachim Lupberger [1,2], Eloi R. Verrier [1,2], Mozhdeh Sojoodi[4], Marine A. Oudot[1,2], François H. T. Duong[1,2,8], Ricard Masia[9], Lan Wei[4], Christine Thumann[1,2], Sarah C. Durand[1,2], Victor González-Motos[1,2], Danijela Heide[5], Jenny Hetzer[5], Shigeki Nakagawa[3], Atsushi Ono [10], Won-Min Song [11], Takaaki Higashi[12], Roberto Sanchez[13], Rosa S. Kim[14], C. Billie Bian[11], Karun Kiani[15,16], Tom Croonenborghs[15,16,17], Aravind Subramanian[15], Raymond T. Chung[18], Beate K. Straub[19], Detlef Schuppan[20,21], Maliki Ankavay[22], Laurence Cocquerel[22], Evelyne Schaeffer[23], Nicolas Goossens[24], Anna P. Koh[3], Milind Mahajan[11], Venugopalan D. Nair [25], Ganesh Gunasekaran[26], Myron E. Schwartz [26], Nabeel Bardeesy [27], Alex K. Shalek [16,28,29], Orit Rozenblatt-Rosen[16,31], Aviv Regev [16,30,31], Emanuele Felli[1,2,7], Patrick Pessaux[1,2,7], Kenneth K. Tanabe[4], Mathias Heikenwälder [5], Catherine Schuster [1,2], Nathalie Pochet [15,16], Mirjam B. Zeisel [1,2,32], Bryan C. Fuchs [4,33✉], Yujin Hoshida [3✉] & Thomas F. Baumert [1,2,7✉]

Chronic liver disease and hepatocellular carcinoma (HCC) are life-threatening diseases with limited treatment options. The lack of clinically relevant/tractable experimental models hampers therapeutic discovery. Here, we develop a simple and robust human liver cell-based system modeling a clinical prognostic liver signature (PLS) predicting long-term liver disease progression toward HCC. Using the PLS as a readout, followed by validation in nonalcoholic steatohepatitis/fibrosis/HCC animal models and patient-derived liver spheroids, we identify nizatidine, a histamine receptor H2 (HRH2) blocker, for treatment of advanced liver disease and HCC chemoprevention. Moreover, perturbation studies combined with single cell RNA-Seq analyses of patient liver tissues uncover hepatocytes and HRH2[+], CLEC5A[high], MARCO[low] liver macrophages as potential nizatidine targets. The PLS model combined with single cell RNA-Seq of patient tissues enables discovery of urgently needed targets and therapeutics for treatment of advanced liver disease and cancer prevention.

A list of author affiliations appears at the end of the paper.

Progressive liver fibrosis, caused by viral (hepatitis B and C viruses [HBV and HCV]) or metabolic (alcohol and non-alcoholic steatohepatitis [NASH]) etiologies frequently leads to highly lethal cirrhosis and hepatocellular carcinoma (HCC), a leading cause of cancer death globally[1]. Advanced liver fibrosis has been shown to be the key risk factor for HCC in NASH with no approved treatment options[2]. Moreover, in hepatitis C patients with advanced fibrosis, HCC risk persists despite viral cure[3]. HCC incidence and death rates have sharply increased compared to other forms of cancer[1]. Given the extremely high prevalence of advanced liver fibrosis and cirrhosis, which are estimated to affect ~1–2% of the global population[2], there is a major unmet medical need for chemoprevention of liver disease progression toward cancer development. While chemoprevention has the potential to significantly impact the prognosis of patients with chronic diseases by reducing lethal complications, the development of effective chemopreventive drugs for advanced liver diseases has been a daunting task as evidenced by the absence of approved therapies[3].

Previously, we have identified a pan-etiology 186 gene clinical prognostic liver signature (PLS) in diseased liver tissues robustly predicting liver disease progression and carcinogenesis in multiple patient cohorts[4–8], as well as animal models[8,9]. While the PLS has been robustly associated with liver disease progression and clinical risk of HCC development, efficient and high-throughput identification of candidate compounds for treatment of advanced liver disease and HCC chemoprevention has been hampered by the absence of tractable model systems and the complex cell circuits of disease biology.

In this work, we describe a simple and robust human cell culture system that models the clinical PLS in an inducible and reversible manner and apply it for drug discovery. By combining our system and validation in rodent and patient-derived models, we discover nizatidine, a histamine receptor H2 (HRH2) blocker, for treatment of advanced liver disease and HCC chemoprevention.

## Results

### Development of a liver cell-based system recapitulating the clinical PLS in cell culture.
To establish a liver cell-based system modeling the clinical PLS, we used the hepatoma-derived Huh7.5.1 cell line differentiated to acquire a hepatocyte-like phenotype (Huh7.5.1$^{dif}$)[10], as shown by global transcriptome profile by 10 days of culture with dimethyl sulfoxide (DMSO; Supplementary Fig. 1). Metabolism- and coagulation-related pathways, the major functions of mature hepatocytes, were gradually induced in hepatocyte-like cells, indicating a shift of the global transcriptome from a malignant to a hepatocyte-like profile. As chronic hepatitis C is one of the well-established inducers of the PLS in patients[5,6], Huh7.5.1$^{dif}$ cells were subsequently infected with HCV strain Jc1 (Fig. 1a, b). Compared to mock-infected cells, persistent viral infection increases the expression poor-prognosis genes and suppresses expression of the good-prognosis genes, leading to the induction of the poor-prognosis pattern of the clinical PLS (Fig. 1c and Supplementary Fig. 2, the full PLS gene list is described in Supplementary Data 1). The global induction or suppression of the PLS signature was then determined by gene set enrichment analysis (GSEA) using mock-infected cells as reference, and results were presented as simplified heatmaps (Fig. 1d). The PLS induction is viral titer- and time-dependent, with full development at day 7 post infection, and virus specific, as shown by infection with highly purified FLAG-tagged HCV (HCV Jc1E2$^{FLAG}$; Fig. 1d and Supplementary Figs. 2 and 3). Moreover, HCV treatment by interferon alpha-2a (IFNα-2a) or direct-acting antivirals (DAAs) partially reversed the PLS (Fig. 1d), corroborating its causal link with persistent HCV infection. ScRNA-Seq analyses revealed correlation of HCV viral load and the PLS

induction in Huh7.5.1$^{dif}$ cells (poor-prognosis genes: $R = 0.5099$, $p = 0.0008$; good-prognosis genes: $R = -0.3751$, $p = 0.0171$, Pearson correlation), further supporting the causality of HCV infection for the induction of the poor-prognosis PLS (Fig. 1e, f).

Along with the induction of the PLS, transcriptome profile of HCV-infected Huh7.5.1$^{dif}$ cells showed that a range of metabolic pathways involved in physiological functions of hepatocytes were suppressed. These results suggest that the PLS induction is accompanied by altered metabolic functions of hepatocytes (Supplementary Fig. 3 and Supplementary Data 2). In contrast, we observed a gradual induction of immune response-related pathways, growth factor signaling pathways and enhanced cell cycle, which are all known to be involved in liver fibrogenesis and carcinogenesis[3]. Taken together, these data indicate that HCV infection of Huh7.5.1$^{dif}$ cells models pathway dysregulation observed in human liver disease.

### Infection with hepatitis viruses and metabolic injuries in cell-based models induce the clinical PLS similar to patient cohorts.
Since there are shared mechanisms leading to hepatocarcinogenesis across the main liver disease etiologies[11], we investigated whether the PLS can be similarly induced in cell-based models by other chronic viral infections or metabolic injuries. To model the PLS for HBV infection, we applied another cell system based on HepG2 cells engineered to overexpress sodium taurocholate cotransporting polypeptide (HepG2-NTCP cells), a cell entry factor for HBV[12]. Persistent HBV infection induced the poor-prognosis PLS (Fig. 1g, and Supplementary Figs. 2 and 4a), however, less robustly than HCV infection. The specificity of the PLS induction in Huh7.5.1$^{dif}$ cells was confirmed by absent induction of the PLS using other hepatotropic viruses, such as hepatitis D virus (HDV), which requires coinfection of HBV to induce liver cirrhosis/cancer, hepatitis E virus (HEV), and dengue virus (DENV), not related to liver cancer (Fig. 1g, and Supplementary Figs. 2 and 4a, b). Ethanol and free fatty acid (FFA) exposure, which cause alcoholic liver disease and induce NASH, respectively, robustly induced the PLS in Huh7.5.1$^{dif}$ cells alone or cocultured with LX2 stellate cells. Interestingly, hypoxia, that promotes fibrogenic liver disease[13], induced expression of the PLS poor-prognosis genes, whereas nutrient starvation did not result in the induction of the PLS (Fig. 1g). The PLS induction appears therefore specific for the two cancer-causing viruses (HBV and HCV) and the two key metabolic injuries (alcohol and dietary FFA) responsible for the fibrogenic/carcinogenic liver diseases in patients.

The clinical relevance of the PLS in the cell-based model (from now referred as cPLS for "cell culture-derived PLS") was confirmed by similar transcriptomic dysregulation in cell culture and the diseased liver of clinical cohorts, with corresponding liver disease etiologies (Fig. 1h, i). A highly significant similarity was observed for PLS induction between the cells treated with the respective etiological agents and the liver tissue of patients with poor prognosis or severe disease manifestation. These data suggest that upon the etiological insults, transcriptional dysregulation observed in the cell culture models mimics gene expression changes in tissues of patients with advanced liver disease.

Previous studies had already suggested that certain cancer cell-based models share signaling pathways with different cell types[14] (e.g., NIH LINCS program, https://clue.io/). Since we observed the PLS induction in monolayer cell culture lacking multicell-type tissue architecture, we investigated whether the cPLS system models transcriptomic dysregulations not only in liver parenchymal but also non-parenchymal cells (NPCs). Interestingly, an analysis of the expression of the PLS genes in the main cell compartments within patient liver tissues[15] revealed that the PLS poor-prognosis genes are most prominently expressed in NPCs,

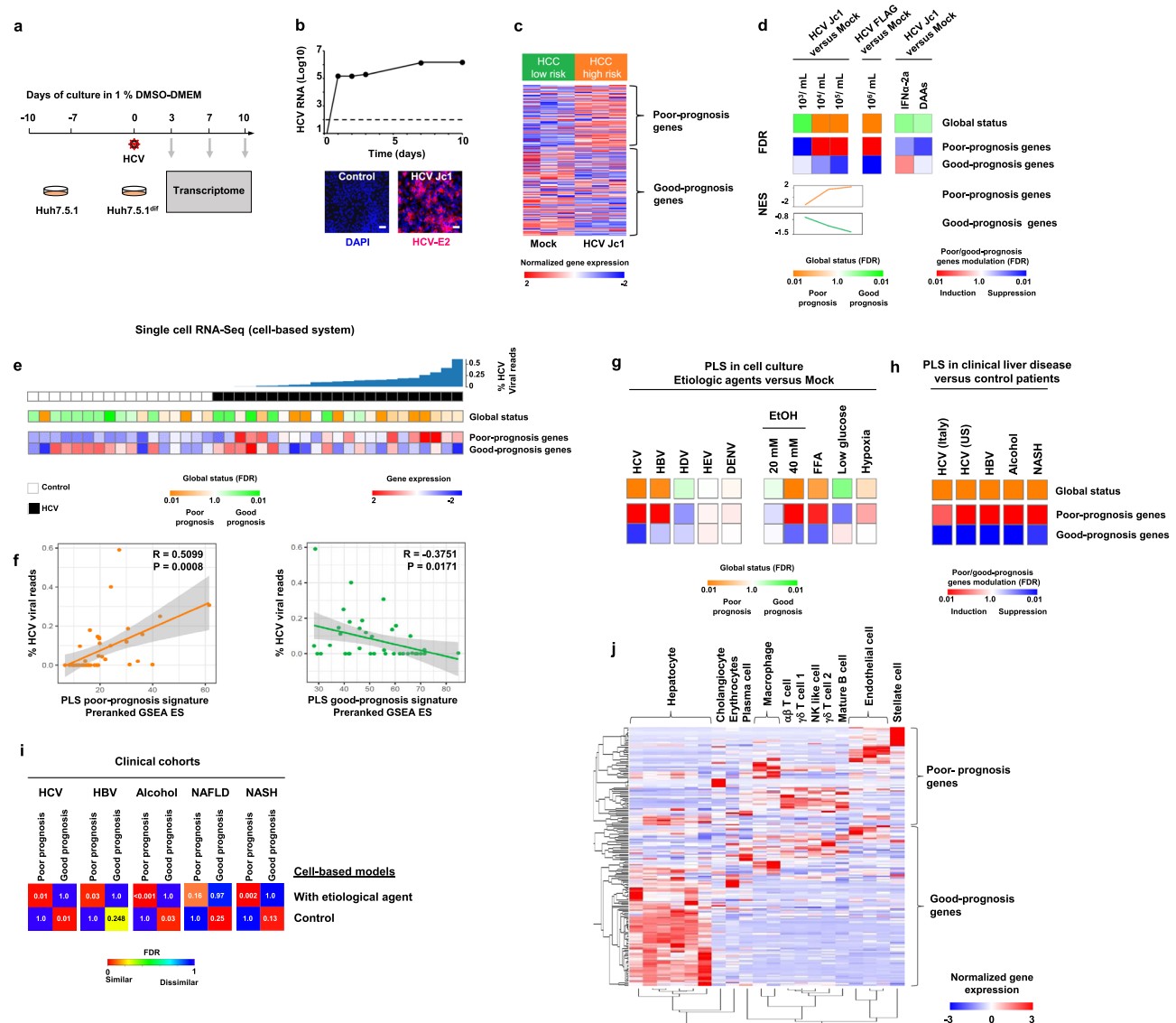

**Fig. 1 Modeling the clinical prognostic liver signature (PLS) in a cell-based system. a** Experimental approach. **b** Analysis of HCV infection by qRT-PCR (log10 of relative RNA quantity normalized to *GAPDH* mRNA; mean ± SD, $n = 3$). Immunodetection of HCV E2 protein 10 days post infection. Scale bar: 50 μm. **c** PLS assay in Huh7.5.1[dif] cells. Heatmap shows expression of poor- and good-prognosis genes (186 gene signature). Results are representative of one out of three independent experiments performed in triplicate. **d** Analysis of the PLS after HCV Jc1 infection and antiviral treatment. PLS induction was determined by GSEA analysis using "mock" non-infected cells as reference. Simplified heatmaps show PLS global status and PLS poor- and good-prognosis gene expression. Results from two experiments performed in triplicate are shown. FDR false discovery rate, NES normalized enrichment score. **e** scRNA-Seq profiling reveals HCV-dependent induction of the PLS in Huh7.5.1[dif] cells. Heatmaps show: PLS global status and modulation of the poor-prognosis (FDR = 0.013) and good-prognosis genes (FDR = 0.016). White: non-infected cells, $n = 17$; black: infected cells, $n = 23$. **f** Enrichment scores of PLS gene modulation for their correlation with the HCV viral load measured at the single-cell level (Pearson correlation tests). **g** PLS assays in the cell-based models. Cell-based models were infected with different viruses or exposed to metabolic cues. HBV infection was performed in HepG2-NTCP cells and free fatty acids (FFA) treatment was performed in Huh7.5.1[dif] cells cocultured with LX2. Results are from three independent experiments. **h** PLS in clinical liver tissues from patient cohorts. HCV-related cirrhosis (Italy, $n = 216$ GSE156546; US, $n = 145$ GSE541027), HBV-related liver cancer ($n = 199$, GSE14520), alcoholic hepatitis ($n = 22$, GSE28619), and NASH ($n = 72$, GSE49541). Induction of the PLS was assessed by comparing diseased tissues with non-diseased tissues. **i** Induction of the PLS in cell-based systems was compared to liver transcriptome profiles from clinical cohorts using Subclass Mapping. **j** PLS deconvolution showing mean gene expression of the poor- and the good-prognosis genes in the main cell compartment within the liver. The scRNA-Seq dataset was extracted from GSE115469[15]. Source data are provided as a Source data file.

whereas the expression of the PLS good-prognosis genes is higher in hepatocytes (Fig. 1j). Next, we analyzed the pathways dysregulated in cirrhotic liver tissues at single-cell level, using recently published scRNA-Seq data from five cirrhotic livers compared to five healthy livers[16]. We compared these pathways with the pathways dysregulated in cPLS model. We observed that the PLS expressing cells share dysregulated pathways with

epithelial cells mainly consisting of hepatocytes and with some NPCs, including immune cells (Supplementary Fig. 5a, b). Together, these data demonstrate that the cPLS system captures molecular pathways of epithelial cells, such as hepatocytes, as well as certain pathways shared between hepatocytes and NPCs.

ScRNA-Seq and flow cytometry analyses show heterogeneity of the cPLS model on a cellular level (Fig. 1e and Supplementary

Fig. 5c–e). To further investigate cell heterogeneity, we assessed the diversity across individual HCV-infected Huh7.5.1[dif] cells, according to *KRT18* expression levels, a marker of hepatocyte. While the sample size was limited, the diverse *KRT18* expression across the cells enabled a robust comparative analysis. We observed no biased expression of the poor- and good-prognosis-associated genes, according to the *KRT18* expression levels, suggesting that there is no differentiation of the HCV-infected Huh7.5.1[dif] cells to form either parenchymal- or NPC-like subpopulation with differential expression of the poor- or good-prognosis-associated PLS genes (Supplementary Fig. 5f). Then, we assessed similarity of global transcriptome between each cell and the major liver cell types in human cirrhotic livers. Interestingly, all Huh7.5.1[dif] cells showed similar magnitude of resemblance to both parenchymal and NPC cell types irrespective of the *KRT18* expression levels and the HCV viral load (Supplementary Fig. 5g). These data support that HCV infection, as an example of etiologic agent to induce the PLS, does not result in cellular heterogeneity with regard to induction of the parenchymal- or NPC-associated PLS genes, i.e., the cPLS system at least partially models the transcriptional program of both parenchymal and NPC liver cell types at comparable level across the Huh7.5.1[dif] cells.

**Screening of computationally prioritized compounds in the cell-based model identifies nizatidine as a candidate chemoprevention agent.** We next hypothesized that the cPLS system enables high-throughput screening of compounds that reverse the PLS poor-prognosis status to a good-prognosis status, and therefore could halt progression of liver disease. To test this hypothesis, we first performed a transcriptome-based in silico drug screening, using the PLS as a query in the chemogenomic database connectivity map (cmap, www.broadinstitute.org/cmap) and LINCS database (https://clue.io/)[17,18], and computationally prioritized 54 candidate compounds (cmap: negative enrichment score with *p* value < 0.05; LINCS: connectivity score < − 90, see "Method"; Supplementary Tables 1 and 2). Next, we further narrowed down the list to 25 compounds currently approved for long-term clinical use, and/or at late stage of clinical development without severe adverse effects (LiverTox database (http://livertox.nlm.nih.gov). Due to its simplicity and robustness, we used persistent HCV infection to induce the cPLS poor-prognosis status. The EGFR inhibitor erlotinib served as a positive control for the PLS reversion[9]. To simplify the protocol, reduce the screening costs and thus increase the clinical translatability of the assay, we used a bioinformatically streamlined version of the PLS comprising 32 genes bioinformatically defined and validated in multiple patient cohorts in previous studies[7,8] (Supplementary Data 1). We identified 11 small molecules that significantly reversed the poor-prognosis PLS (hit rate 44%; Fig. 2a and Supplementary Fig. 6). Importantly, the effect of the compounds on the PLS was not due to an antiviral effect (Fig. 2a). The identification of known drugs with a chemopreventive effect, such as pioglitazone or resveratrol[19,20], confirms the validity of the system. Interestingly, the top candidate compounds target different and complementary pathways, suggesting that several signaling pathways contribute to the induction of the PLS.

Next, the transcriptional targets of each identified compound were extracted from the LINCS database (the top 100 genes most modulated by the compounds) and tested for their enrichment during HCV induction of the poor-prognosis cPLS at the single-cell level (Fig. 2b). We observed a specific induction of each compound-associated signature during HCV infection, correlated with the cPLS induction (Fig. 2b). Importantly, the hit compounds suppressed their target gene expression,

demonstrating (i) their specificity of action and (ii) that they reverse the PLS by modulating these specific targets (Fig. 2c).

The PLS-reversing compounds with highest statistical significance (FDR < 0.05) included captopril, an angiotensin-converting enzyme inhibitor used as antihypertensive, and nizatidine, a HRH2 antagonist used as gastric acid reducer (Fig. 2a). Retrospective studies have demonstrated a therapeutic effect of captopril and other inhibitors of the renin–angiotensin system for liver fibrosis and inflammation[21]. These data indicate that perturbation of the cPLS translates into a clinical therapeutic effect.

Interestingly, our screen uncovered the MEK inhibitors CI-1040, PD0325901, and Selumetinib as candidate compounds. MEK is a protein kinase involved in the Ras/Raf/MEK/ERK signaling pathway, which has been shown to play a functional role in HCC biology through regulation of apoptosis, cell cycle, cell migration, differentiation, and proliferation. MEK inhibitors are currently in clinical investigation for HCC treatment[22]. Since nizatidine has a well-established clinical safety profile and has not been explored in detail for treatment of chronic liver disease and HCC prevention, we further explored nizatidine and its targets in the liver.

**Nizatidine reverts the cPLS by targeting the HRH2/cAMP/CREB signaling pathway.** HRH2, the nizatidine target, is a member of the G-protein-coupled receptor family widely expressed in the gastrointestinal tract that mediates its activity through cAMP[23]. To investigate the mechanism of action of nizatidine in our cell-based system, we first confirmed the expression of HRH2 in Huh7.5.1[dif] cells by immunofluorescence analyses (Fig. 3a). Induction of the poor-prognosis cPLS by histamine, the natural ligand of HRH2, with reversal by nizatidine demonstrated the functional activity of HRH2 in Huh7.5.1[dif] cells, and confirmed the functional impact of HRH2 in the modulation of the cPLS (Fig. 3b). These results were corroborated by an increase of intracellular cAMP level following histamine treatment, reverted by nizatidine (Fig. 3c). Furthermore, incubation of the cells with 8-CPT-cAMP, a cAMP analog, resulted in induction of the poor-prognosis cPLS (Fig. 3b). Treatment of the cells with H89, a specific PKA inhibitor, reversed the poor-prognosis cPLS, demonstrating the involvement of the cAMP/PKA axis in the induction of the cPLS (Fig. 3b). Of note, Huh7.5.1[dif] cells express histidine decarboxylase (HDC), the enzyme catalyzing histamine production from histidine (Fig. 3d). HDC expression was increased upon HCV infection, suggesting that these cells produce more histamine for stimulation of histamine receptors in stress conditions (Fig. 3d). These data are in line with studies showing that hepatoma cells express histamine receptors and produce histamine[24,25].

Increase of intracellular cAMP is known to activate the cAMP response element binding (CREB) protein family, which has key functions in cell survival, growth, inflammation, and differentiation[26]. We focused on CREB1 and CREB5, two key members of the CREB family found to be overexpressed in many solid tumors, including HCC[26–28]. HCV infection resulted in an increase of CREB1 phosphorylation and of CREB5 expression, both reversed by nizatidine (Fig. 3e and Supplementary Fig. 7), suggesting a role of these transcription factors in the induction of the poor-prognosis cPLS. Loss-of-function studies using CRISPR/Cas9 revealed that knockout (KO) of *CREB5* reversed the poor-prognosis cPLS, verifying CREB5 as a key mediator of the cPLS (Fig. 3f and Supplementary Fig. 7). The finding that *CREB5* KO results in robust reversal of the PLS is most likely due to the fact that CREB5 is a transcription factor activated by several pathways, which mediate expression of PLS genes. Furthermore,

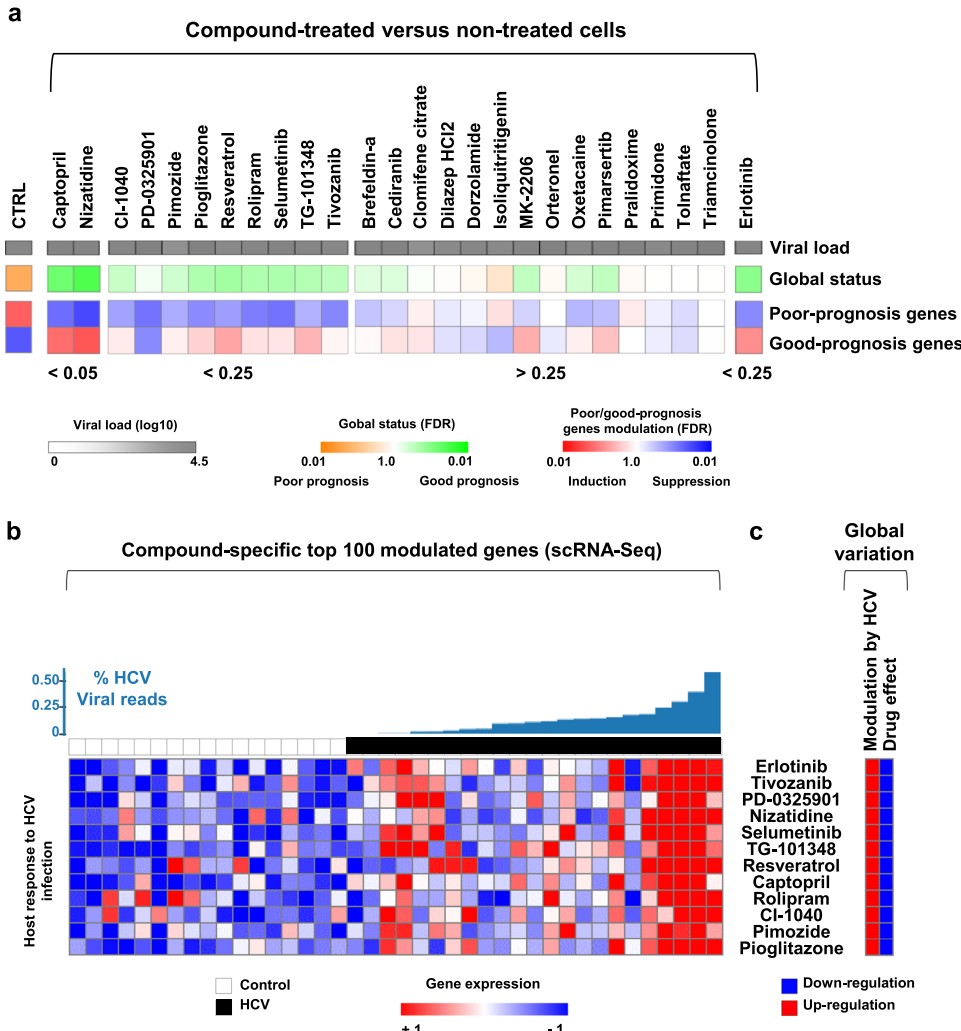

**Fig. 2 A high-throughput screen with the clinical PLS as a readout identifies nizatidine as candidate compound for treatment of chronic liver disease and HCC prevention. a** Effect of computationally selected compounds on the PLS. Simplified heatmaps show the HCV viral load after drug treatment (log 10), PLS global status, and PLS poor- and good-prognosis gene expression. PLS induction was determined by GSEA analysis using "mock" non-infected cells as reference control. Reversal of the PLS was assessed by comparing "HCV-infected cells treated with compounds" versus "HCV-infected cells (CTRL)". The compounds were then ranked depending on the FDR (significant FDR < 0.25). Results from drug screening performed in triplicate are shown. **b** Screen hit target genes modulated by HCV infection at the single-cell level. Gene signatures comprising the targets of each specific compound were extracted from the LINCS database and tested for their enrichment in association with the HCV viral load. Heatmaps show the mean expression of the leading-edge genes for the significantly enriched signatures ($p < 0.05$) in each of the single cells (columns), row normalized as z-score. The single cells are ordered by infection status (white: non-infected cells, $n = 17$; black: infected cells, $n = 23$), and by increasing HCV viral load (blue bar plot on top). **c** Global modulation of the compound-specific gene signatures by HCV infection and drug treatment. Source data are provided as a Source data file.

CREB5 itself is a transcriptional activator for expression of several PLS genes[5,26]. Collectively, our results demonstrate that the histamine/HRH2/cAMP/CREB signaling pathway is a mediator of the cPLS poor-prognosis status (Fig. 3g).

To validate the key findings in primary cells, we established a 3D multicellular spheroid model derived from patient tissues in which the 186 gene poor-prognosis PLS is robustly, and significantly induced by FFA exposure and reversed by nizatidine treatment similar as in the cPLS cell line model (Fig. 3h and Supplementary Table 3). Interestingly, the induction and reversal of the PLS poor-prognosis status in this model were associated with the level of HRH2 expression in individual patients (Fig. 3i). Moreover, the HRH2 ligand histamine robustly induced the poor-prognosis PLS status, which was reversed by nizatidine (Fig. 3j). Collectively, these data in primary cells of patient-derived liver spheroids confirm that histamine is a mediator of the PLS

poor-prognosis status and the effect of PLS reversal in primary tissues is dependent on HRH2 expression—the nizatidine target.

**Nizatidine reduces liver fibrosis and cancer development in two animal models for chronic liver disease and HCC by impairing hepatocyte HRH2/CREB5 signaling.** Next, we validated the therapeutic effect of nizatidine in a rat model of progressive fibrotic liver disease and HCC based on repeated low-dose injections of diethylnitrosamine (DEN). This model closely mimics global transcriptome dysregulation in human cirrhosis at risk of HCC, including the PLS induction[8,9]. Treatment with nizatidine had a marked effect on hepatocarcinogenesis: nizatidine-treated animals showed a reduced number of liver tumor nodules (Fig. 4a–c). Moreover, nizatidine tends to decrease expression of the Ki-67 protein in the liver, a marker associated with proliferation of mammalian cells (Fig. 4c). Importantly,

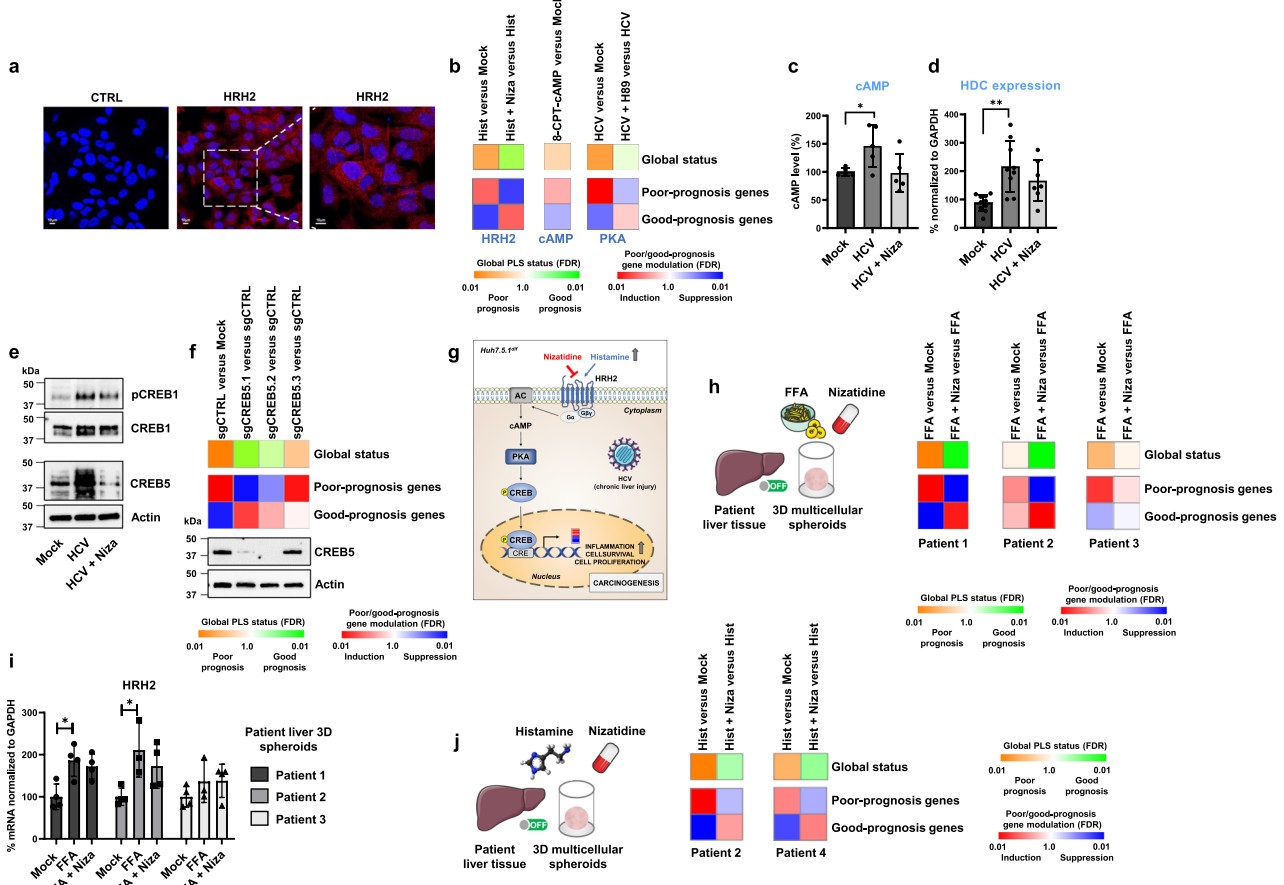

**Fig. 3 Nizatidine reverses the cPLS by inhibiting the HRH2/CREB signaling pathways. a** Detection of HRH2 in Huh7.5.1dif cells by immunofluorescence. HRH2 is shown in magenta (Alexa Fluor™ 647) and nuclei in blue (DAPI). CTRL = cells incubated with AF647-labeled secondary antibody. Scale bar: 10 μm. **b** PLS assessment upon perturbation of the HRH2/cAMP/PKA axis. PLS induction was determined by GSEA analysis using "Mock" non-treated cells as reference. Results from two experiments performed in triplicate are shown. Simplified heatmaps show PLS global status and PLS poor- and good-prognosis gene expression. FDR false discovery rate. **c** Intracellular levels of cAMP were assessed by ELISA. Results from two experiments are expressed in % ±SD compared to Mock. ($n = 5$; *$p < 0.05$, unpaired $t$ test). **d** Expression of histidine decarboxylase (HDC) in Huh7.5.1dif cells analyzed by qRT-PCR. Results from three independent experiments are expressed in % ±SD compared to Mock cells and normalized to *GAPDH* mRNA. ($n = 7$: **$p < 0.01$, two-tailed Mann–Whitney $U$ test). **e** Nizatidine inhibits HCV-mediated CREB1 activation and decreases CREB5 expression. Western blot analysis of phospho-CREB1 (pCREB1) (Ser133), total CREB1, and total CREB5. NT non-treated. Results are representative of one out of two experiments (see Supplementary Fig. 7). **f** CREB5 is a driver of the HCV-induced cPLS. *CREB5* KO was performed as described in "Method". PLS induction was determined by GSEA analysis using "Mock" non-infected cells as reference. Reversal was assessed by analyzing sgCREB5 VS sgCTRL. **g** Activation of the HRH2 signaling pathways during chronic liver injury. AC: adenylate cyclase, CRE CREB response element. **h** Nizatidine reverts the FFA-induced poor-prognosis PLS in culture of patient-derived multicellular spheroids (three patients without history of chronic liver disease). PLS induction was determined by GSEA analysis using "Mock" non-treated spheroids as reference. Simplified heatmaps show PLS global status and PLS poor- and good-prognosis gene expression. **i** HRH2 expression measured by qRT-PCR in healthy multicellular spheroids. Each experiment shows mean ± SD in percentage compared to Mock spheroids ($n = 4$), *$p < 0.05$, unpaired $t$ test. **j** Nizatidine reverts the histamine-induced poor-prognosis PLS in culture of patient-derived multicellular spheroids. Source data are provided as a Source data file.

nizatidine does not induce hepatic toxicity as shown by the measurement of hepatic enzymes (Fig. 4d). Decrease of gamma-glutamyl transferase (γGT) activity and of total bilirubin (TBIL) level showed that nizatidine restores normal liver function assumedly by halting liver fibrosis progression (Fig. 4d). Indeed, nizatidine significantly reduced liver fibrosis measured by collagen proportionate area (CPA), α smooth muscle actin (αSMA) staining, quantitative analysis of hydroxyproline, and expression of genes mediating fibrogenesis (*Acta2*, *Col1a1*, *Tgfb-1*, and *Timp1*; Fig. 4e, f). Taken together, these data demonstrate that nizatidine prevents fibrotic liver disease progression and cancer development in vivo. To better elucidate the antifibrotic/anticancer effect of nizatidine, we performed RNA-Seq analysis on rat liver tissues. Nizatidine reversed the poor-prognosis status of the

PLS (Fig. 4g, h) as observed in the cell-based model, suppressed several pathways mediating fibrogenesis and carcinogenesis (i.e., TGFβ and KRAS signaling) and improved metabolic pathways required for the physiological liver function (i.e., lipid metabolism; Fig. 4g).

We next investigated the expression of nizatidine targets in rat liver tissues. Immunostaining analysis showed that HRH2 and CREB1 phosphorylation are markedly increased in hepatocytes after DEN-mediated liver injury and suppressed by nizatidine, suggesting that HRH2 expression and CREB1 activation are associated with carcinogenic status of the liver (Fig. 4i, j). Decrease of cAMP levels after nizatidine treatment confirmed target engagement in vivo (Fig. 4j). In addition, RNA-Seq analyses revealed that expression of *Prkacb* (encoding PKA

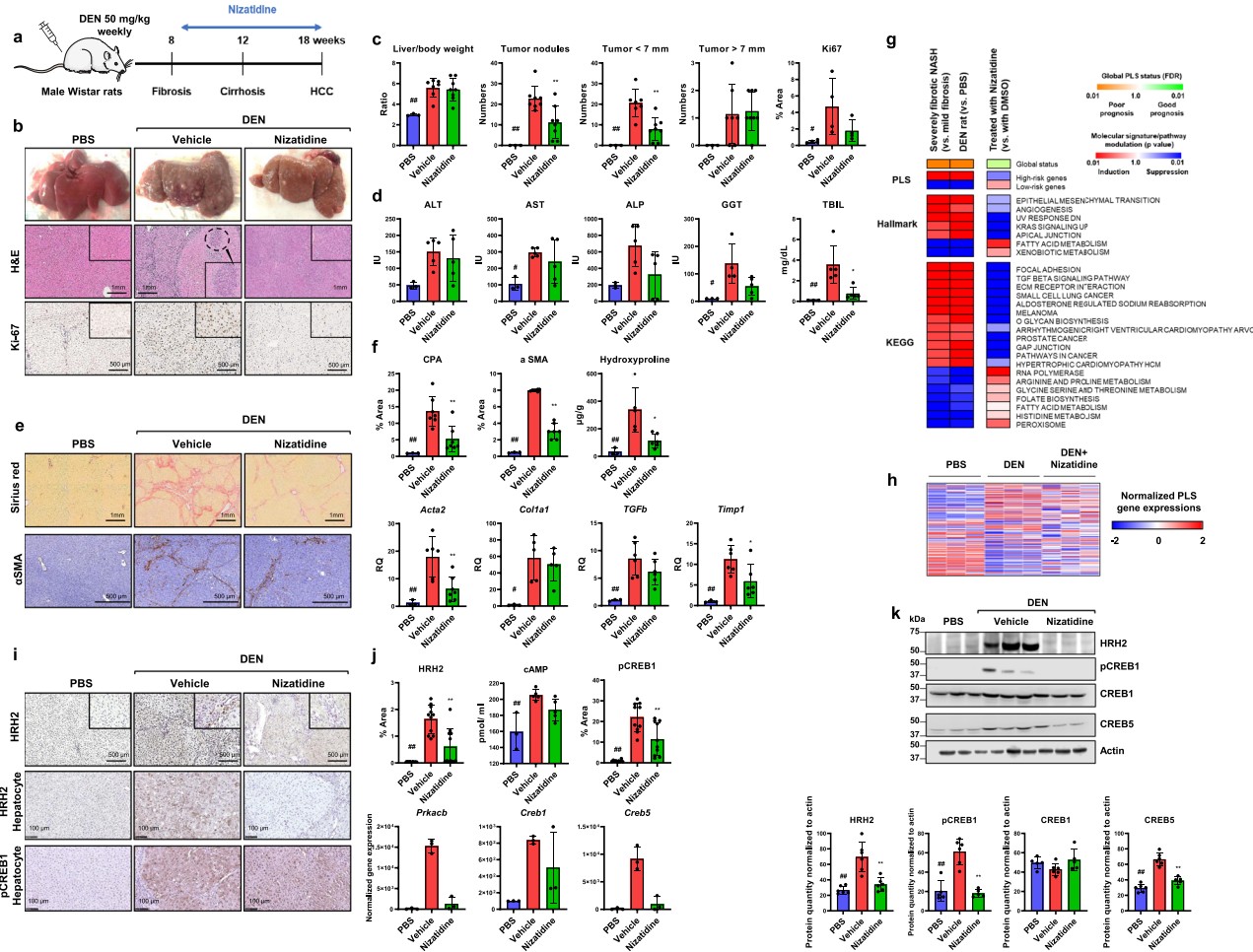

**Fig. 4 In vivo proof-of-concept of nizatidine for treatment of liver fibrosis progression and HCC chemoprevention. a–d** Nizatidine alleviates carcinogenesis in cirrhotic livers. **a** DEN-injured male Wistar rats received vehicle control or nizatidine for 10 weeks. PBS, $n = 3$; DEN + vehicle, $n = 7$; DEN + nizatidine, $n = 8$. **b** Representative morphometric analysis (hematoxylin and eosin (H&E)) of liver slices are shown (original magnification ×5). The number and size of macroscopic tumors were reported in **c**. Surrogates used for tumor burden include liver/body weight ratio and number of surface tumor nodules. Cell proliferation was assessed by the IHC evaluation of the proliferation marker Ki-67 in liver tissues. **d** Measurement of serum transaminases (alanine aminotransferase (ALT), aspartate aminotransferase, (AST) alkaline phosphatase (ALP), gamma-glutamyl transferase (γGT) activity, and of total bilirubin are shown. **e, f** Nizatidine efficiently reduces fibrosis. Liver specimens were stained with Sirius red and fibrosis stage was evaluated through quantitative digital analysis of whole-scanned liver sections (collagen proportional area, CPA). αSMA staining, hydroxyproline quantification and fibrotic gene expression are shown. **g** RNA-Seq analysis of the liver from three animals subjected to either vehicle or nizatidine treatment. Commonly dysregulated pathways in both severely fibrotic NASH liver (compared to mildly fibrotic NASH liver) and DEN-treated rat liver, and pathways reversed by nizatidine were determined by GSEA. **h** Heatmaps show the mean expression of the 186 gene signature (z-scores of log2-normalized data). Gene expression was normalized according to six different housekeeping genes. **i–k** Expression of nizatidine target genes. **i** Liver specimens were stained with anti-HRH2 or anti-pCREB1 antibodies. Original magnification ×10 or ×20 (hepatocytes). **j** HRH2 expression and CREB1 phosphorylation were assessed by quantification of IHC. Levels of cAMP were assessed by ELISA. *Creb1*, *Creb5*, and *Prkacb* expression were extracted from RNA-Seq analyses. **k** Western blot analysis of HRH2, phosphorylated (Ser 133) CREB1, total CREB1, and CREB5. Beta-actin was used as a loading control. Quantification of western blot intensities (arbitrary units) was performed using image J software. All graphs show mean ± SD. # denotes $p < 0.05$ and ## denotes $p < 0.01$ comparing PBS-treated rat to DEN + vehicle. * denotes $p < 0.05$ and ** denotes $p < 0.01$ comparing DEN + vehicle to DEN + nizatidine (One-way ANOVA, followed by Tukey's multiple comparisons tests). Source data are provided as a Source data file.

catalytic subunit beta) and *Creb5* was increased, following DEN liver injury, with reversal by nizatidine (Fig. 4j). Similar results were found for protein expression as shown by immunoblot studies, corroborating our findings (Fig. 4k and Supplementary Fig. 8).

To address a direct relationship of nizatidine treatment and HRH2 expression, we treated cell lines with nizatidine and quantified HRH2 expression after treatment. As nizatidine had no effect on HRH2 expression in cell lines or 3D spheroids (Fig. 3i and Supplementary Fig. 9), we assume that the observed downregulation of HRH2 in nizatidine-treated animals is most

likely not due to a direct effect of nizatidine on HRH2 expression, but rather due to the improvement of liver disease.

We then investigated the effect of nizatidine on liver disease progression in a mouse model of NASH/fibrosis/HCC induced by a choline-deficient, L-amino acid-defined, high-fat diet (CDAHFD) and a single dose DEN injection. This diet-based model closely recapitulates the course of disease development, histological appearance, and dysregulation of fatty acid metabolism in human NASH progressing to fibrosis and HCC[29]. As observed in the rat model, nizatidine markedly reduced liver fibrosis and HCC (Supplementary Fig. 10a–d). An increase of

HRH2 and CREB5 expression and CREB1 phosphorylation in the liver upon injury, reversed by nizatidine treatment, was further confirmed in liver tissues from treated animals (Supplementary Figs. 10f–h and 11). Finally, similar to the rat model, nizatidine reversed the poor-prognosis PLS in the livers of treated mice (Supplementary Fig. 10i–j).

**Genetic loss-of-function studies confirm a functional role of HRH2 in hepatocarcinogenesis.** To confirm that the observed inhibitory effects of nizatidine on hepatocarcinogenesis are mediated by its target HRH2, we performed genetic loss-of-function studies in animal and cell-based models. First, we performed in vivo KO studies using hydrodynamic tail vein injection (HTVI) in a mouse model of liver cancer. HTVI enables efficient delivery of plasmid DNA into the liver of adult mice, in particular into hepatocytes[30]. In our model, hepatocarcinogenesis and *Hrh2* KO were induced by injection of three plasmids, one expressing the Cas9 endonuclease and sgRNAs (targeting *Tp53, Hrh2* or a non-targeting sgRNA control), the transposon plasmid expressing Kras$^{G12D}$ allowing constitutive expression of the Kras mutant[31], and the third plasmid expressing a transposase that integrates the transposon vector randomly into the genome of transduced cells (Fig. 5a, b). *Hrh2* KO robustly and significantly ($p < 0.01$) improved mouse survival compared to control animals (median survival: CTRL 28d, *Hrh2* KO 33.5d; Fig. 5b). Moreover, *Hrh2* KO decreased the liver tumor burden (Fig. 5c, d) and significantly decreased cell proliferation (Fig. 5e).

To investigate the functional role of HRH2 on the phenotype of hepatoma cells, we engineered Huh7.5.1 *HRH2* KO and CTRL KO cell lines. *HRH2* KO hepatoma cells showed decreased cell proliferation (Fig. 5f, g), and increased sensitivity to oxidative stress and apoptosis compared to cells expressing a sgCTRL (Fig. 5h and Supplementary Fig. 12). A decrease of Huh7.5.1 cell proliferation was also observed when HRH2 expression was knocked down by RNAi (Fig. 5i–k). Moreover, *HRH2* KO impaired the full induction of the poor-prognosis status of the cPLS compared to CTRL cells (Fig. 5l). Collectively, these genetic loss-of-function studies suggest that HRH2 plays a functional role in hepatocarcinogenesis, and that the biological effects of nizatidine on liver disease progression are likely mediated by HRH2. Nevertheless, our data do not exclude that additional mechanisms or targets are at play.

**Nizatidine improves liver inflammation and immune surveillance by targeting liver macrophages.** Increasing evidence has shown that the interplay between hepatocytes and the surrounding microenvironment plays an important role in liver disease progression and hepatocarcinogenesis. In particular, the recruitment of inflammatory immune cells in the chronically injured tissue is crucial for driving fibrosis and HCC[32]. Given the role of histamine and histamine receptors in the regulation of immune responses and inflammation[33], we hypothesized that targeting HRH2 using nizatidine may modulate liver inflammation and immunity.

First, we assessed the effect of nizatidine on immune cell infiltration in livers from nizatidine-treated animals. We observed a decrease of the Ly6G, CD68, and F4/80 cell markers, indicating a reduced infiltration of neutrophils and monocytes/macrophages in livers from treated animals (Supplementary Fig. 13). Moreover, an increase of CD3- and CD8-specific markers after nizatidine treatment indicated a recruitment of CD8$^+$ T lymphocytes (LT; Supplementary Fig. 13), which could contribute to the anticarcinogenic effect of nizatidine[34]. In the NASH-fibrosis mouse model the reduction of inflammatory immune cell recruitment

leads to a decrease in the expression of *interleukin 6* (*Il6*) and *chemokine ligand 2* (*Ccl2*), two pro-inflammatory cytokines contributing to liver disease progression and HCC development[34] (Supplementary Fig. 13).

To uncover the immune cells targeted specifically by nizatidine, we first analyzed target expression in the healthy human liver using scRNA-Seq. ScRNA-Seq of human liver cell atlas[15,35] showed that HRH2 is highly expressed in liver macrophages (Fig. 6a and Supplementary Fig. 14a). Immunofluorescence analysis and qRT-PCR performed on isolated macrophages and hepatocytes confirmed that HRH2 is expressed in macrophages, as well as hepatocytes with higher levels in macrophages (Supplementary Fig. 14b–d). Macrophage populations include a wide spectrum of phenotypes from the classically activated pro-inflammatory macrophages (M1) to alternatively activated immunoregulatory macrophages (M2)[36]. Characterization of macrophage marker expression revealed that HRH2-expressing macrophages express high level of *C-type Lectin Domain Family 5 member A* (*CLEC5A*) and low level of *MAcrophage Receptor with COllagenous structure* (*MARCO*) and *CD163 Molecule Like 1* (*CD163L1*) markers, indicating that HRH2-expressing liver macrophages are mainly pro-inflammatory (Fig. 6a and Supplementary Fig. 14a)[15,37].

To investigate the functional effects of nizatidine on liver immune cells, we isolated CD45$^+$ leukocytes from liver tissues of patients with advanced liver disease and HCC, and treated the immune cell population with nizatidine or vehicle control (Fig. 6b and Supplementary Fig. 15). We then analyzed the effect of the compound by scRNA-Seq using the SORT-seq technology similar to our previously described liver cell atlas pipeline[35,38]. Cell clustering is based on similar transcriptomic profiles. The different immune cell types were identified using specific cell markers (Fig. 6c, d)[35]. As indicated by their marked shift within the *t*-SNE map the cell population with highest change in gene expression corresponds to liver macrophages (CD45$^+$ *MAF BZIP Transcription Factor B* (*MAFB*)$^+$ cells; Fig. 6c, d). Interestingly, the majority of nizatidine-responding macrophages have a pro-inflammatory phenotype, as demonstrated by high level of *CLEC5A* expression and low levels of *CD163L1* and *MARCO* expression (Fig. 6e). Nizatidine treatment decreased *CLEC5A* expression (Fig. 6e), suggesting that nizatidine modulates the phenotype of inflammatory macrophages. Furthermore, we observed that nizatidine decreases expression of the *Receptor Sialic-Acid-Binding Ig-like Lectin 10* (*Siglec-10*) in macrophages (Fig. 6e), a recently uncovered immune checkpoint shown to inhibit effector functions of immune cells in cancer[39].

GSEA analysis showed that nizatidine suppresses the pro-inflammatory M1 signature and partially induces the immunoregulatory M2 signature (Fig. 6f). Macrophages exhibit remarkable plasticity and can evolve in different subpopulations with atypical or intermediate profiles sharing characteristics of more than one population[36]. In accordance with this observation, we show that nizatidine suppresses pro-inflammatory and pro-fibrogenic responses (i.e., TNFα, IL2, and IL6 signaling pathways), while augmenting antigen processing and presentation and IFNγ response, suggesting a shift from pro-inflammatory to an "atypical" immunoregulatory profile (Fig. 6g). Moreover, we observed a strong suppression of the c-Myc pathway expression, a major actor involved in the polarization of macrophages towards a typical M2 profile[40]. Gene-level analysis also showed that nizatidine treatment induces a decrease in neutrophil/monocyte chemoattractant cytokine expression (i.e., CXCL5, CCL2, and CCL5), as well as a suppression of pro-fibrotic/tumorigenic soluble factor expression (i.e., IL6, IL1β PDGF, MMP9, and TNFα; Fig. 6h)[34]. A direct effect of nizatidine treatment on

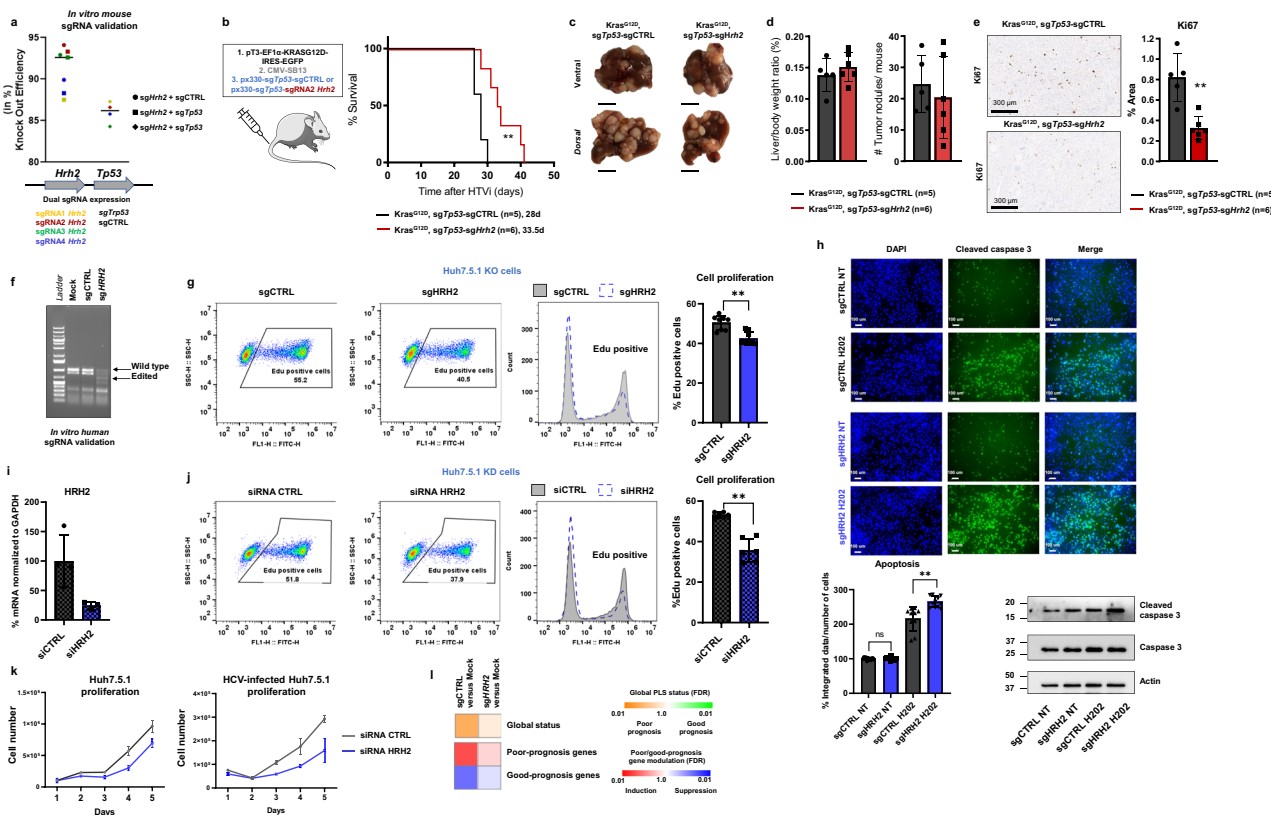

**Fig. 5 Genetic loss-of-function studies confirm a functional role of HRH2 in hepatocarcinogenesis. a** Validation of sgRNA targeting mouse *Hrh2* and *Tp53*. KO efficacy was assessed by TIDE analysis. Plasmid constructs are described in method. **b** *Hrh2* (sg*Hrh2*) and *Tp53* (sg*Tp53*) KO and *Kras* constitutive expression (*KRAS^{G12D}*) or CTRLs were induced by injecting different plasmids constructs by HTVI in C57BL/6 males. Graph shows survival curves of injected mice. Median survival from injection to death is shown. Log-rank Matel-Cox test. **$p < 0.01$. **c** Pictures of representative livers. Scale bars, 1 cm. **d** Liver weight body ratio and total number of tumor nodules per mouse are shown (mean ± SD; CTRL mice, $n = 5$; *Hrh2* KO mice, $n = 6$). $p > 0.05$ (two-tailed Mann–Whitney $U$ test). **e** Cell proliferation was assessed by the IHC evaluation of Ki-67 in liver tissues (mean ± SD; CTRL mice, $n = 5$; *Hrh2* KO mice, $n = 6$). Scale bar, 300 μM. **$p < 0.01$ (two-tailed Mann–Whitney). **f**, **g** *HRH2* KO decreases cancer cell proliferation in cell culture. **f** *HRH2* KO validation at genetic level using T7 endonuclease assay. **g** Effect of *HRH2* KO on cancer cell proliferation. EdU incorporation assay by FACS showing % ±SD of proliferative EdU-positive cell from three independent experiments in CTRL and *HRH2* KO cells ($n = 8$; **$p < 0.01$, two-tailed Mann–Whitney $U$ test). **h** Effect of *HRH2* KO on cancer cell apoptosis induced by oxidative stress ($H_2O_2$). Cleaved caspase 3 is shown in green. Nuclei were counterstained in blue (DAPI). Scale bar, 100 μM. Graph shows integrated cleaved caspase 3 intensity/total cell number from two independent experiments ($n = 12$; **$p < 0.01$, two-tailed Mann–Whitney $U$ test) measured using Celigo Cytometer. Western blot analysis of cleaved- and total caspase 3 is shown. **i–k** Effect of *HRH2* knockdown on cancer cell proliferation. **i** siRNA efficacy was assessed by measuring mRNA by qRT-PCR. **j** EdU incorporation assay by FACS showing % ±SD of proliferative EdU-positive cell from three independent experiments in cell transfected with siCTRL and siHRH2 ($n = 6$; **$p < 0.01$, two-tailed Mann–Whitney $U$ test). **k** Cell proliferation was assessed daily in Huh7.5.1 transfected with a siCTRL or a siHRH2 by cell counting (TC20 Automated Cell Counter). Two representative and independent experiments are shown. **l** Full cPLS induction is impaired by *HRH2* KO. PLS induction was determined by GSEA analysis using "Mock" non-infected cells as reference. Source data are provided as a Source data file.

pro-inflammatory cytokine expression was confirmed in THP1-derived macrophages with similar effects of other HRH2 blockers, as well as patient-derived Kupffer cells and patient-derived tumor associated macrophages, (Supplementary Fig. 16). To address the functional role of HRH2 in macrophages by an additional experimental approach, we generated a *HRH2* KO THP1-derived macrophage cell line and assessed their functional phenotype. Interestingly, the *HRH2* KO resulted in a similar modulation of pro-inflammatory cytokines as treatment with nizatidine (Supplementary Fig. 16d, e). These data suggest that at least a part of the observed immunomodulatory effect of nizatidine in THP1-derived macrophages is mediated through HRH2 and that macrophage HRH2 likely contributes to the therapeutic effect of nizatidine in vivo.

**HRH2 and downstream target expression predicts mortality in patients with chronic liver disease.** Next, we sought to validate

the clinical relevance of the target pathways in liver disease progression and hepatocarcinogenesis in different clinical cohorts. Transcriptome profiling revealed that HRH2 and CREB5 expression were increased in viral hepatitis and NASH, confirming involvement of the pathway in disease progression across the major etiologies (Fig. 7a, b). Increase of HRH2 expression was further confirmed at protein level in liver tissues from patients with chronic liver diseases and HCC (Supplementary Fig. 17). HRH2 protein was detected both in hepatocyte and liver immune cells, with the highest protein expression levels in immune cells. Moreover, we observed that high HRH2 expression is associated with poorer overall survival in cirrhotic patients at risk for HCC (Fig. 7c). Expression of the downstream effector CREB5 was also associated with poorer overall survival and higher incidence of HCC recurrence after curative tumor resection (Fig. 7d). Collectively, these results support a functional relevance of the HRH2 pathway in pan-etiology liver disease progression and hepatocarcinogenesis in patients.

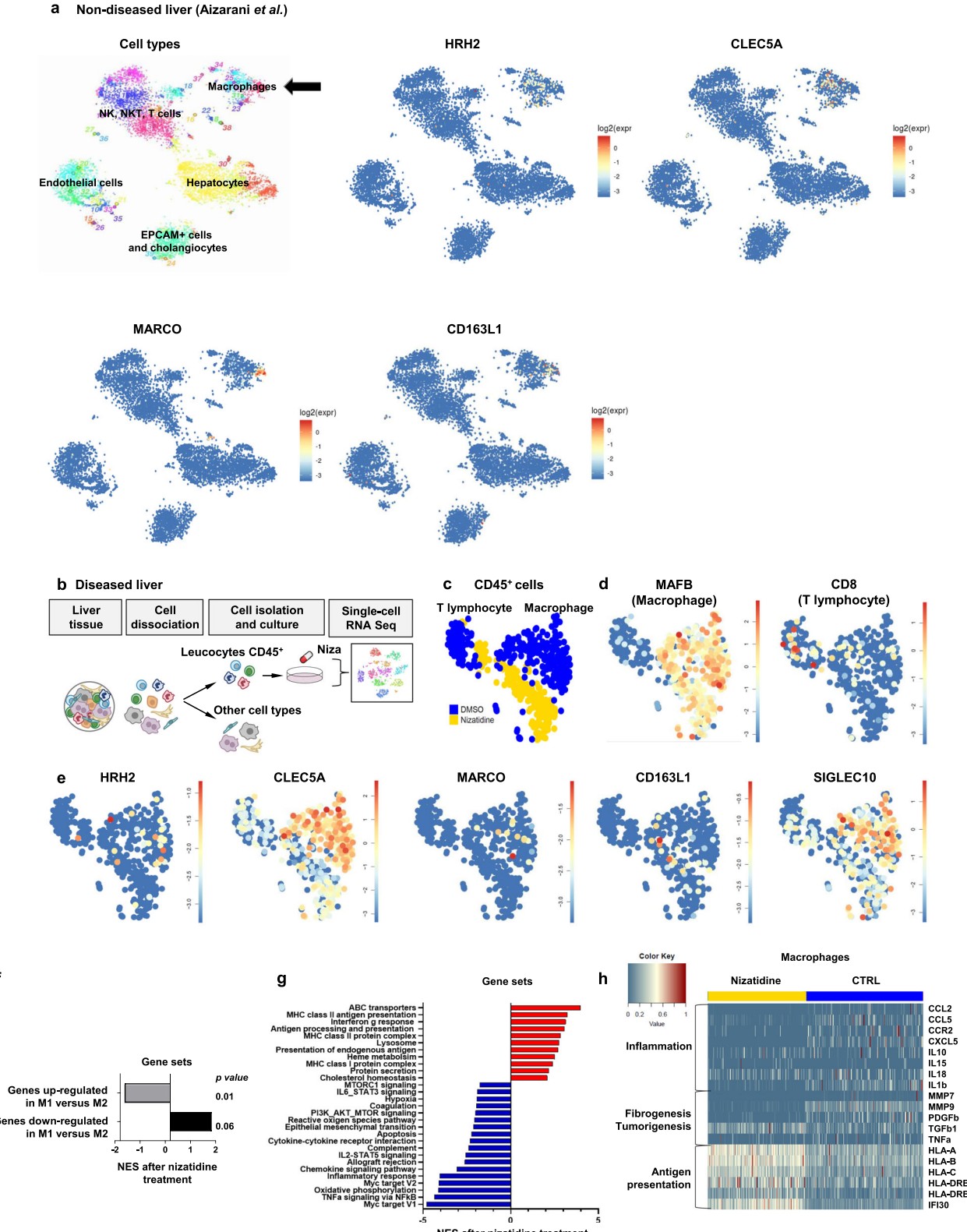

**Nizatidine reverts the PLS poor-prognosis status and decreases HCC cell viability in ex vivo patient-derived models.** Finally, we aimed to investigate the potential clinical efficacy of nizatidine in advanced liver disease. First, we performed an organotypic ex vivo culture of diseased, non-tumoral human liver tissues, preserving multicell-type tissue architecture, including liver NPCs [8,41]. Nizatidine treatment reverted the poor-prognosis PLS (Fig. 8a), suggesting a therapeutic effect on liver disease progression and HCC risk. To assess whether nizatidine may also contribute to a direct effect on emerging HCC, we investigated nizatidine effect in 3D patient-derived HCC tumorspheroids. In contrast to other 3D culture systems, HCC spheroids include the

**Fig. 6 ScRNA-Seq analyses of patient liver tissue uncover pro-inflammatory liver macrophages as nizatidine target. a** *t*-SNE map of single-cell transcriptomes from normal liver tissue of donors without history of chronic liver disease highlighting the main liver cell compartments. Data extracted from ref. [35]. Cells sharing similar transcriptome profiles are grouped by clusters and each dot represents one cell. Expression *t*-SNE map of *HRH2*, *CLEC5A*, *CD163L1*, and *MARCO* are shown. The color bar indicates Log2-normalized expression. **b**–**e** Perturbation of gene expression by nizatidine in liver tissue from patient with chronic liver disease and HCC identifies liver macrophages as therapeutic target. **b** Experimental approach. CD45+ leukocytes from patient liver tissue were enriched by flow cytometry and were treated with nizatidine or vehicle control (DMSO). Single cells were sorted and analyzed as described[35]. **c** *t*-SNE map of single-cell transcriptomes showing control (blue) and nizatidine-treated cells (yellow), **d** the *t*-SNE map indicating the main cell compartments (MAFB: macrophages, CD8: CD8+ T lymphocytes). **e** Expression *t*-SNE map of HRH2, macrophage markers, and Siglec-10 are shown. The color bar indicates log2-normalized expression. **f**–**h** GSEA for differentially expressed genes between nizatidine-treated and CTRL macrophages depicted in **d**. **f** Normalized enrichment score (NES) of genes related to macrophage activation (classical M1 vs alternative M2). **g** NES of the pathways significantly enriched after nizatidine treatment (FDR ≤ 0.05). **h** Expression heatmap of differentially expressed genes in individual nizatidine- and control-treated macrophages. Each row representing a single cell. Markers of inflammation, fibrogenesis/cancer, and antigen presentation are shown. All genes are normalized by row from their own min to max (Log2 fold; *p* value ≤ 0.05). Source data are provided as a Source data file.

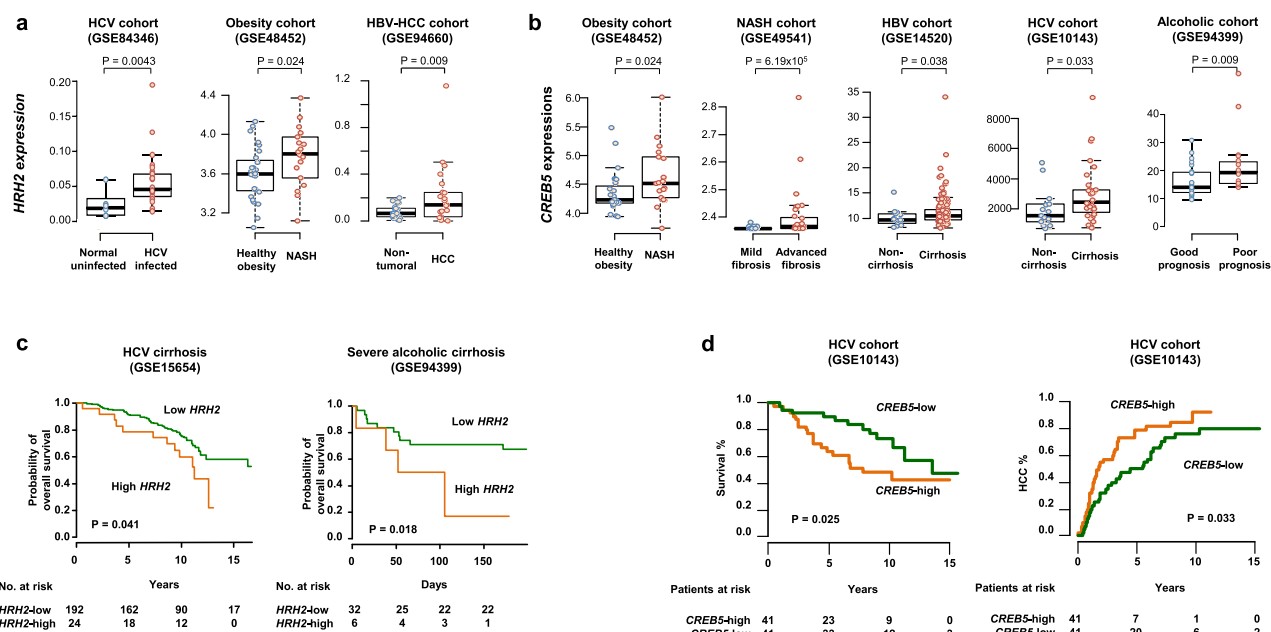

**Fig. 7 Expression of the liver HRH2/CREB5 pathway is associated with chronic liver disease progression and hepatocarcinogenesis in patients. a**, **b** HRH2 (**a**) and CREB5 (**b**) expression in liver tissues of clinical cohorts with various liver disease etiologies. In box and whisker plots, boxes represent the 75th and 25th percentiles, the whiskers represent the most extreme data points within interquartile range × 1.5, and the horizontal bar represents the median. Open circle indicates actual observation for each sample. Exact *p* values are indicated for each panel (Wilcoxon's rank-sum test for all panel excepted for GSE94660, paired Wilcoxon's rank-sum test). GSE84346: normal uninfected, *n* = 6, HCV infected, *n* = 40; GSE48452: healthy obesity, *n* = 27; NASH, *n* = 18; GSE94660, paired samples, *n* = 21; GSE48452: healthy obesity, *n* = 27; NASH, *n* = 18; GSE49541: mild fibrosis, *n* = 40, advanced fibrosis, *n* = 32; GSE14520: non-cirrhosis, *n* = 14, cirrhosis, *n* = 185; GSE10143: non-cirrhosis, *n* = 15, cirrhosis, *n* = 32. GSE94399: good prognosis, *n* = 23; poor prognosis, *n* = 15. **c** Higher expression of HRH2 in patient liver biopsies is associated with poor-prognosis and decreased probability of overall survival. Exact *p* values are indicated for each panel (log-rank test for comparisons of Kaplan–Meier survival). **d** Higher expressions of CREB5 in the adjacent livers were significantly associated with decreased survival and tumor recurrence after curative resection among patients with HCC. Exact *p* values are indicated for each panel (log-rank test for comparisons of Kaplan–Meier survival). Source data are provided as a Source data file.

tumor microenvironment, allowing establishment of cell–matrix and cell–cell contacts between hepatocytes and NPCs, including macrophages[42]. While nizatidine had no effect on primary human hepatocyte (PHH) viability in line with its safe clinical profile (Fig. 8b), treatment with the compound resulted in a significant decrease of tumorspheroid viability in four out of six patients, independently of the etiology, including a pronounced effect on tumors that did not respond to sorafenib (Fig. 8c and Supplementary Table 3). Nizatidine-induced alteration of liver macrophage gene expression (Fig. 6) was associated with a robust ex vivo response to the drug in the same patient (Fig. 8c, HCV HCC 2).

Collectively, these functional data suggest that targeting HRH2 may have clinical efficacy in patients with advanced chronic liver disease.

## Discussion

There is an unmet medical need for experimental systems modeling human disease-specific gene expression to understand disease biology and enable disease-specific drug discovery. Here, we addressed this need by the development of a simple and robust liver cell-based system that models gene expression of patients with carcinogenic liver diseases. Our findings demonstrate that a clinical prognostic signature can be experimentally modeled in a cell-based model. Overall, the cPLS model offers unique opportunities to discover compounds for chronic liver disease treatment and HCC chemoprevention across the distinct liver cancer etiologies, in a fast-track high-throughput screening format. Moreover, using detailed computational and functional analyses, we demonstrate that the cPLS model system recapitulates global transcriptomic dysregulation in hepatocyte-like and

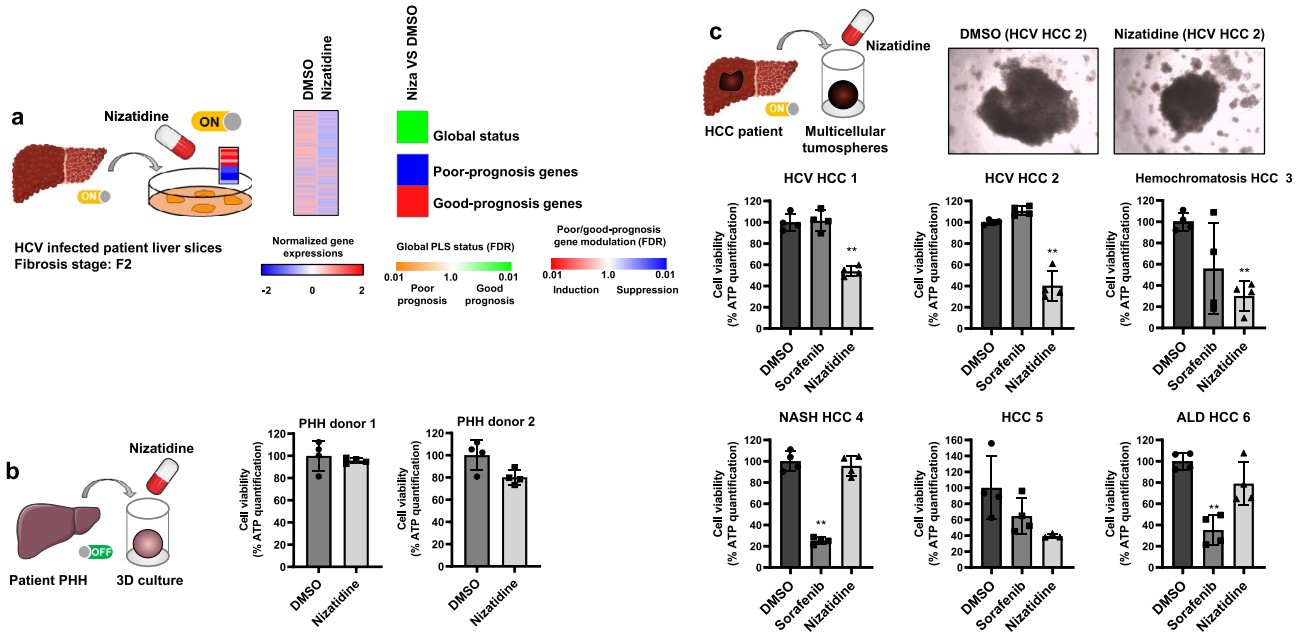

**Fig. 8 Proof-of-concept for therapeutic impact of nizatidine in patient-derived tissues and cell culture models. a** Nizatidine reverts the poor-prognosis PLS in culture of patient-derived tissue that were surgically resected from five patients diagnosed with chronic hepatitis C (HCV). Detailed PLS gene expression profiles: heatmaps show the mean expression of the 186 gene signature. PLS was determined by GSEA analysis using DMSO treated tissues as reference. Simplified heatmaps show: (top) the classification of PLS status as poor (orange) or good (green) prognosis; (bottom) the significance of induction (red) or suppression (blue) of poor- or good-prognosis genes. FDR false discovery rate. **b** Absent effect on cell viability in PHH, assessed 4 days after nizatidine treatment in 3D culture. Each experiment shows mean ± SD in percentage compared to DMSO treated cells ($n = 4$). **c** Nizatidine decreases HCC cell viability in a 3D patient-derived tumorspheroid model. HCC spheroids were generated from patient HCC tissues with different etiologies. Cell viability was assessed 4 days after treatment by measuring ATP levels. Each experiment shows mean ± SD in percentage compared to DMSO treated spheroids ($n = 4$). *$p < 0.05$; **$p < 0.01$, unpaired $t$ test. The pictures show representative image of patient-derived tumorspheroids (magnification ×40). NASH nonalcoholic steatohepatitis, ALD alcoholic liver disease. Source data are provided as a Source data file.

non-hepatocyte cell types, including macrophages, which supports versatile utility of our system to screen and identify compounds targeting various liver cell types. These findings are in line with studies indicating that modeling of cellular signaling is not necessarily restricted by cell type and tissue lineage[8,14,18,43]. Indeed, such cell-based in vitro models (e.g., NIH LINCS program) have revealed that cellular signaling and physiology are often shared across multiple tissues and cell types[14]. As any cell culture model, the cPLS platform only partially recapitulates the cell circuits of the major liver cell types in the diseased liver in a simplified manner, and therefore the platform should be used for compound or target screening followed by validation in multicell-type experimental systems, such as patient-derived spheroids, as well as rodent models of advanced liver disease as demonstrated in this study. However, the robust induction of the clinical PLS demonstrates that the model is useful to identify compounds reversing the poor-prognosis status of the signature (Fig. 2) or pathways involved in the generation of the signature (Fig. 3). Induction of the PLS in authentic patient tissue-derived spheroids and animal models and its reversal by the top-scored compound (Fig. 3h) confirms the validity of the cell line-based model. Whereas the spheroid cPLS model is closer to the patient allowing validation of compounds and mechanisms, the limited availability of patient-derived tissues preclude its application for compound screening.

The therapeutic effect of nizatidine was mediated by two complementary mechanisms:

(1) Hepatocytes/parenchymal cancer cells: we demonstrate that the HRH2/CREB5 signaling pathway is perturbed in hepatocytes during liver injury. Moreover, we show that

HRH2 is involved in cancer cell proliferation (Fig. 5). The clinical impact of these findings was confirmed by an increased expression of HRH2 and CREB5 in HCC patients, and a significant association of HRH2/CREB5 expression with patient overall survival and enhanced HCC risk in several clinical cohorts (Fig. 7). Our discovery is consistent with studies demonstrating that CREB5 overexpression in parenchymal cells is associated with clinical tumor recurrence, metastasis, poor prognosis, and overall survival[27,28]. CREB5 is a transcription factor belonging to the CREB protein family that regulates diverse cellular responses, including proliferation, survival, and differentiation. Upregulation of CREB protein can transform normal parenchymal cells into tumor cells through aberrant activation of downstream pathways, such as growth factor receptor (i.e., EGFR) and cytokine/JAK/STAT pathways[26].

(2) Liver macrophages: scRNA-Seq analyses in patient tissue uncovered HRH2[+] CLEC5A[high] MARCO[low] liver macrophages as another candidate target cell for nizatidine (Fig. 6). Clinical and experimental evidences have shown that macrophages play a key role in liver fibrosis progression[16]. Furthermore, certain macrophage subpopulations enhance tumor progression by impairing cytotoxic LT CD8[+] immune responses and are emerging as a target in cancer therapy[39]. We show that nizatidine enhances pathways mediating antigen processing and presentation in macrophages, which is in line with the observed increase of LT CD8[+] recruitment after nizatidine treatment in vivo (Supplementary Fig. 13). LT CD8[+] responses in human HCC correlate with improved overall survival, longer relapse-free survival, and diminished disease progression[34].

Furthermore, we observed that nizatidine treatment resulted in decreased expression of macrophage Siglec-10 (Fig. 6). Activation of Siglec-10 by its ligands in macrophages induces a "don't eat me signal", which blocks phagocytosis of transformed cells and has very recently been shown to contribute to immune evasion[39]. Therefore, it is conceivable that the decrease of Siglec-10 expression by nizatidine may contribute to a restoration of anticancer immunity. The two target cells and mechanisms are most likely linked by hepatocyte–macrophage cross talk.

While our genetic loss-of-function studies demonstrate a functional role for HRH2 in liver disease cell circuits and hepatocarcinogenesis, and our functional data in hepatocyte and macrophage cell lines suggest a class effect of HRH2 inhibitors, we cannot exclude that nizatidine may exert the observed biological effects also via targets other than HRH2.

More than 80% of HCCs develop in fibrotic or cirrhotic livers, suggesting an important role of liver fibrosis in the carcinogenesis[1,2]. In patients with chronic viral infection and NASH, the risk of HCC gradually increases as liver fibrosis progresses[3,44]. Targeting fibrosis-associated carcinogenesis may be relevant to prevent HCC development. By improving liver inflammation, fibrosis, and anticancer surveillance, HRH2-targeting compounds provide a therapeutic candidate approach for patients with fibrotic liver disease at risk for HCC and will guide future optimization of refined HRH2-targeting liver disease therapies. The excellent safety profile combined with robust therapeutic efficacy at doses, which are achieved in patients treated with HRH2 antagonists[45], suggest rapid translatability of the approach.

## Methods

**For reagents, primers, antibodies, and other resources, see Supplementary Table 4**

*Human subjects.* Human liver tissues were obtained from liver disease patients undergoing liver resection with informed consent from all patients for deidentified use at the Center for Digestive and Liver Disease of the Strasbourg University Hospitals University of Strasbourg, France (DC-2016-2616 and RIPH2 LivMod IDRCB 2019-A00738-49, ClinicalTrial NCT04690972) or at Mount Sinai Hospital, New York City, NY (HS13-00159). The protocols were approved by the local Ethics Committee of the University of Strasbourg Hospitals and Mount Sinai Hospital, respectively. All material was collected during a medical procedure strictly performed within the frame of the medical treatment of the patient. Informed consent is provided according to the Declaration of Helsinki. Detailed patient information and informed consent procedures are implemented by the Strasbourg University Hospital Biological Resources Center (HUS CRB). Patients were given an information sheet, which outlines that their leftover biological material (liver resection and blood samples) that was collected during their medical treatment is requested for research purposes. All patients received and signed an informed consent form in order to provide authorization or refuse the use of their biological samples (protocols DC-2016-2616 and RIPH2 LivMod IDRCB 2019-A00738-49 ClinicalTrial NCT04690972). The patients maintain the right to withdraw their consent at any time and to request the destruction of their biological material, which is strictly respected. While there was clinical descriptive data available, the identity of the patients was protected by internal coding. A brief summary of patient characteristics (diagnosis and treatments) is provided in Supplementary Table 3. Spheroids (Fig. 3) were generated from liver tissue from patient without history of chronic liver disease. Tumorspheroids (Fig. 8) were generated from tumoral tissues from HCC patients. The PLS was analyzed in clinical liver tissues from HCV-related cirrhosis (Italy, n = 216 GSE15654 (ref. [6]); US, n = 145 GSE54102[7]), HBV-related liver cancer (n = 199, GSE14520)[46], alcoholic hepatitis (alcohol; n = 22, GSE28619)[47], and NASH (n = 72, GSE49541)[48] cohorts. Induction of the PLS in cell-based systems was compared to liver transcriptome profiles from published clinical cohorts of HCV-related cirrhosis (n = 145, GSE54102)[7], HBV-related liver cancer, alcoholic hepatitis, nonalcoholic fatty liver disease (NAFLD; n = 131, GSE31803)[49], and NASH (n = 73, GSE48452)[50], using Subclass Mapping[50].

*Research experiments on live vertebrates.* For rat and mouse models for liver disease and HCC, animals were housed in accordance with the guidelines of the Massachusetts General Hospital Institutional Animal Care and Use Committee (protocol approval numbers 2007N000113 and 2009N000207) and received human care, according to the criteria outlined in the "Guide for the Care and Use of Laboratory Animals" of the National Academy of Sciences. Eight-week-old male Wistar rats

(Charles River Laboratories, Wilmington, Massachusetts) received weekly intraperitoneal injections of 50 mg/kg DEN (n = 18) to induce cirrhosis and liver cancer, or PBS (n = 3), once per week over the course of 18 weeks. After 8 weeks, DEN-injured rats were randomly assigned to receive vehicle control (0.5% methylcelluose; n = 9) or 15 mg/kg nizatidine (n = 9) by oral gavage daily for 9 weeks by a blinded technician. Drug doses were estimated based on body surface area assuming a nizatidine dose of 150 mg for a 60 kg human[51]. Livers were harvested and analyzed in week 18. Male C57Bl/6 mice (Charles River Laboratories, Wilmington, MA) received a single dose of 35 mg/kg DEN at day 15. At 6 weeks of age, mice were subjected to either standard chow (n = 5) or CDAHFD consisting of 60 kcal% fat and 0.1% methionine by weight (n = 16) for a total of 24 weeks. Oral gavage of either vehicle control (n = 8) or nizatidine 15 mg/kg (n = 8) was initiated 6 weeks following the onset of CDAHFD. At the time of sacrifice, animals were anesthetized and sedated. A terminal blood collection was performed by cardiac puncture. Livers were weighted, fixed in formalin, and snap-frozen for further analysis.

For HTVI, C57BL/6 male mice were purchased at an age of 8-weeks old from Janvier Labs and were used for experiments. All experiments were performed in accordance with the German law and the governmental bodies, and with approval from the the Regierungspräsidium Karlsruhe providing the ethical oversight of the study (approval number G39/18). Mice were housed at the German Cancer Research Center (DKFZ; constant temperature of 20–24 °C and 45–65% humidity with a 12-h light–dark cycle) and maintained under specific pathogen-free conditions. For in vivo CRISPR/Cas9 gene editing by HTVI. Groups of six 8–10-week-old male mice received a mix of 5 μg DNA of transposon vector pT3-EF1α-KRASG12D-IRES-EGFP (KRASG12D), 10 μg of either px330-sgTp53-Ctrl (sg-Tp53-sgCtrl) or px330-sgTp53-sgHth2 (sgTp53-sgHrh2), and a 5:1 ratio of transposon to CMV-SB13 transposase-encoding plasmid dissolved in 0.9% NaCl solution and injected in the lateral tail vein, with a total volume corresponding to 10% of body weight in 5–7 s. Once the animals were sacrificed, livers were collected, formalin-fixed and paraffin-embedded, frozen, or embedded in OCT (TissueTek).

*Cells.* Huh7.5.1 are a gift of Dr. F. Chisari (The Scripps Research Institute, La Jolla, CA). HepG2 and THP1 were purchased from ATCC. LX2 were purchased from Merck. Huh7.5.1, HepG2, and human stellate LX2 cells were cultured in Dulbecco's Modified Eagle Medium (DMEM) supplemented with 10% heat-decomplemented fetal bovine serum (FBS), gentamycin (0.05 mg/mL), and nonessential amino acids (complete DMEM) at 37 °C with 5% CO₂. THP1 cells were cultured in RPMI 1640 Medium with GlutaMAX™-I supplement and HEPES, and supplemented with 10% FBS and gentamycin (0.05 mg/mL). NTCP-overexpressing Huh7.5.1-NTCP or HepG2-NTCP cells have been generated by transducing a human NTCP-expressing VSV pseudoparticle (GeneCopoeia). After 3 days, the cells were expanded and selected for NTCP expression with 1.8 μg/mL puromycin[12]. For proliferation arrest and differentiation (Huh7.5.1dif cells), Huh7.5.1 cells were cultured in complete DMEM-supplemented 1% DMSO[10]. All cell lines were certified mycoplasma-free. For perturbation studies in the cell-based system, the dose of nizatidine was chosen according to the nizatidine blood concentrations measured in patients. Plasma concentrations reach between 700–1400 μg/L (2–4 μM) after a 150 mg oral dose and between 1400–3600 μg/L (4–11 μM) after a 300 mg oral dose[45].

*Primary human hepatocyte (PHH) isolation.* PHH were isolated as described[35] from patient-derived liver tissues obtained from surgical resections (Strasbourg University Hospitals, France). Briefly, liver specimens were perfused for 15 min with calcium-free 4-(2-hydroxyethyl)-1-piperazine ethanesulfonic acid buffer containing 0.5 mM ethylene glycol tetraacetic acid (Fluka), followed by perfusion with 4-(2-hydroxyethyl)-1-piperazine ethanesulfonic acid containing 0.5 mg/mL collagenase (Sigma-Aldrich) and 0.075% CaCl₂ at 37 °C for 15 min. Then the cells were washed with phosphate-buffered saline (PBS) and nonviable cells were removed by Percoll® (Sigma-Aldrich) gradient centrifugation. Part of the isolated cells was further separated into PHH and NPCs fractions by an additional centrifugation step (5 min at 50 g). PHH were seeded in collagen type I coated plates (Corning®).

*PLS calculation.* The PLS 186 (complete) or 32 gene (reduced, see below) expression profiling was performed using 250–500 ng total RNA by using either nCounter Digital Analyzer system (NanoString) or the HumanHT-12 beadarray (Illumina) for the time-course experiments (Supplementary Figs. 1 and 3). For full PLS gene list, refer to Supplementary Data 1. PLS gene expression was normalized according to six housekeeping gene expression (Supplementary Data 1), using GenePattern genomic analysis toolkits[52,53]. Detailed PLS gene expression profiles are presented as heatmaps showing the mean expression of the 186 or 32 PLS genes, using Morpheus software (z-scores of log2-normalized data). Induction or suppression of the PLS signature was determined as previously reported by using GSEA, implemented in GenePattern genomic analysis toolkits[52,53]. PLS was always determined by using CTRL cells, CTRL animals, or CTRL patient-derived tissues as references. Results are presented as simplified heatmaps showing the classification of PLS global status as poor or good prognosis, and the significance of induction/suppression of PLS genes (log10 of false discovery rate (FDR) values). Global status corresponds to the difference between good- and poor-prognosis gene expression.

The statistic tests, normalized enrichment score (NES), and FDR values are provided for each PLS experiment (poor- and good-prognosis genes variation and global status) in the Source data file.

The 32 gene signature is a reduced version of the PLS, comprising gene bioinformatically defined and validated in multiple patient cohorts in previous studies[7,8]. The gene signature was bioinformatically reduced from 186 genes to 32 genes (Supplementary Data 1), and implemented in an FDA-approved diagnostic assay platform specifically designed for clinical use (Elements assay, NanoString) as Laboratory Developed Test (LDT). The Clinical Laboratory Improvement Amendments (CLIA) lab implementation of the assay is now underway to provide the assay service soon for clinicians worldwide[54–57] (https://www.utsouthwestern.edu/aboutus/administrativeoffices/technology/development/about/).

**Transcriptomic analyses.** Molecular pathway deregulations were determined in Molecular Signature Database (MSigDB, ver.4.0)[58] using GSEA[59]. Comparison between the cell-based systems and clinical datasets was performed using Subclass Mapping algorithm[50] implemented in GenePattern. Transcriptome profiling of nizatidine-treated rat liver was performed by RNA-Seq. Sequencing libraries were generated using the TruSeq RNA Sample Prep Kit (Illumina) using 200 ng of total RNA, according to the manufacturer's protocol and sequenced, using Illumina NextSeq genome sequencer (Illumina). Data preprocessing and transcript abundance calculation (RPKM: reads per kilo bases of exons for per million mapped reads) were performed using TopHat2.1 and Cufflinks 2.2 software, using the rat reference genome, rn4 (ref. [60]). Rat genes were converted into orthologous human genes using a mapping table obtained from NCBI HomoloGene database (build 68). Induction or suppression of target genes of relevant cellular signaling pathways was assessed by GSEA.

**Statistics and reproducibility.** Individual experiments were reproduced three times in an independent manner with similar results (except otherwise stated). The precise number ($n$) of biologically independent samples used to derive statistics is indicated in the figure legends. The in vitro data are presented as the mean ± SD and were analyzed by the unpaired Student's $t$ test or the two-tailed Mann–Whitney $U$ test as indicated in figure legends, after determination of distribution by the Shapiro–Wilk normality test. $p < 0.05$ (*), $p < 0.01$ (**) were considered statistically significant. Significant $p$ values are indicated by asterisks in the individual figures. The exact $p$ values are provided in the Source data file. The data points are presented with the bar charts. Statistical analyzes were performed with GraphPad Prism 8 software. No statistical analyzes were performed if $n < 3$. For in vivo experiments, the sample size estimate was based on a $p$ value of 0.01 at 90% power assuming a 50% difference in means in tumor burden with 33% standard deviation between control and drug-treated animals. The one-way ANOVA followed by Tukey's post hoc test was used to compare the three groups. $p < 0.05$ (* or #), $p < 0.01$ (** or ##) were considered statistically significant. Number of animals are indicated in figure legends. Some analyses were performed only on representative animals from each group, i.e., RNA-Seq, measurement of serum transaminases and hydroxyproline... The exact $p$ values are provided in the Source data file. For the PLS assay, variation of the poor- and the good-prognosis genes was determined by a NES obtained, using GSEA analysis. Significance of the data was determined by the FDR values. According to GSEA (https://www.gsea-msigdb.org/gsea/index.jsp), results are significant if FDR < 0.25. Results are express as heatmap (log 10 of FDR). All the exact $p$ values, NES and FDR are provided in the Source data file. For clinical data presented in Fig. 7, data were extracted from publicly available data base (https://www.ncbi.nlm.nih.gov/geo/). Number of patient and statistics tests are indicated in figure legend. Representative graphs or heatmaps from independent experiments with similar results are indicated in figure legends. Due to the rarity of patient liver tissues, experiments involving patient-derived spheroids/tumorspheroids were performed one time in quadruplicate.

**HCV infection.** Cell culture-derived HCVcc Jc1 (ref. [61]) and HCVcc Jc1E2[FLAG][62] (genotype 2a/2a) were produced in Huh7.5.1 cells by electroporation of viral RNA. For HCVcc JC1, the cell culture supernatants were harvested every 3 days for a total of 12 days and were concentrated ten times using Vivaspin 20, 10,000 MWCO PES (Sartorius). HCV Jc1E2[FLAG] was purified using anti-FLAG M2 affinity gel (Sigma-Aldrich), according to manufacturer's instructions. In parallel, Huh7.5.1 were subjected to the electroporation protocol without viral RNA, and the supernatants were collected and concentrated in the same conditions (mock-electroporated cells). HCV infectivity was determined by calculating the tissue culture infective dose 50 ($TCID_{50}$). To analyze the PLS induction Huh7.5.1[dif] cells were infected with HCV Jc1 or HCV Jc1E2[FLAG] (TCID $10^6$ infectious particle/mL), for a total of 10 days. Cell culture supernatants from mock-electroporated cells (without viral RNA) or 100 µg/mL of FLAG peptide were used for control experiments. HCV infection was assessed by qRT-PCR of intracellular HCV RNA and immunostaining, using HCV E2-specific AP33 antibody. To assess the effect of IFNα-2a or DAAs on the PLS, Huh7.5.1[dif] cells were infected with HCV Jc1 for 7 days and were incubated for 3 more days (for a total of 10 days) with either 10 IU/mL IFNα-2a, or a combination of 1 nM daclatasvir and 1 µM sofosbuvir (DAA treatment). PLS induction in virus-infected cells was always determined using mock-infected cells as control references.

**Single-cell RNA-Seq profiling of HCV-infected Huh7.5.1[dif] cells.** Huh7.5.1[dif] cells were infected with HCV Jc1E2[FLAG] (TCID50 = $7 \times 10^5$/mL). On day 7, single cells were sorted into 96-well plates. Single cells were lysed in 5 µL TCL buffer (Qiagen) supplemented with 1% 2-mercaptoethanol. HCV RNA was co-amplified with cellular mRNA using a SMART-compatible primer (sequence: 5′-Biotin-AAG-CAGTGGTATCAACGCAGAGT ACTCTGCGGAACCGGTGAGTA-3′). Paired-end 25 bp reads were sequenced for 59 single cells using the SmartSeq2 protocol[63]. For all single cells, reads were aligned to the human hg19 UCSC reference, as well as the HCV Jc1 reference using Tophat[60,64] and gene expression levels were quantified for 21,948 human genes using Cuffquant from Cufflinks2.2 (refs. [60,64]). The 42 single cells with a read alignment rate of at least 50% and with at least 100,000 pairs of aligned reads were normalized using Cuffnorm with default settings, including "geometric" normalization, which normalizes samples based on the median expression level in each sample, thus reporting normalized expression levels (FPKM: fragments per kilo bases of exons for per million mapped reads). Forty out of 42 single cells were submitted for downstream analyses after excluding samples, i.e., one noninfected single cell that was potentially contaminated and one infected single cell with significantly lower liver cell gene detection levels. Induction or repression of PLS poor- and good-prognosis genes, in association with the HCV viral load in scRNA-Seq data was determined using the preranked GSEA module implemented in GenePattern[53,59], with Pearson correlation as the rank metric. The FDR values of the preranked GSEA analyses that are based on the Kolmogorov–Smirnov statistic, specifically, FDR = 0.013 for poor-prognosis and FDR = 0.016 for good-prognosis genes. Enrichment score of poor- and good-prognosis genes with the HCV viral load was determined using Pearson correlation tests and the $p$ value as estimated by the corrcoef function in Matlab.

**HBV and HDV infection.** HBV (genotype D) was purified from the serum of a HBV carrier. Viral particles were concentrated through ultracentrifugation by pelleting over a 30% sucrose cushion with subsequent gradient centrifugation using a 10–45% iodixanol density gradient[12]. HepG2-NTCP cells were infected with purified HBV or control preparation in the presence of 4% polyethylene glycol and maintained in culture for additional 10 days in 2% DMSO-complemented primary hepatocyte maintenance medium (PMM) and HBV infection was assessed by immunodetection of HBsAg[12]. NTCP-overexpressing Huh7.5.1[dif] cells were infected with recombinant HDV in presence of 4% polyethylene glycol and cultured for 10 days in 2% DMSO-complemented PMM[12]. HDV infection was assessed by immunodetection of HDAg using serum-derived anti-HDAg antibodies. PLS induction in virus-infected cells was always determined using mock-infected cells as control references.

**HEV infection.** HEV viral stocks were produced by electroporation of PLC3 cells (human hepatoma cell line) with gt3 Kernow C-1 p6 strain (GenBank accession number JQ679013)-capped RNAs[65]. Transfected PLC3 cells were maintained at 32 °C in medium containing DMEM/M199 (1 v:1 v), 1 mg/mL of lipid-rich albumin (Albumax I), 1% of nonessential amino acids, and 1% of pyruvate sodium (Life Technologies). Supernatant of confluent T75 flasks of HEV producing cells was harvested, centrifuged for 10 min at 800 r.p.m., and used to infect Huh7.5.1[dif] cells in 1% DMSO-complemented DMEM. At 10 days post infection, infected Huh7.5.1[dif] cells were either used for total RNA extraction or processed for indirect immunofluorescence. Briefly, cells were fixed with 3% of paraformaldehyde (PFA) for 20 min, washed twice with PBS, and permeabilized for 5 min with cold methanol and then with 0.5% Triton X-100 for 30 min. Cells were incubated in PBS containing 10% goat serum for 30 min at room temperature (RT) and stained with the 1E6 (Millipore, 1/800) anti-ORF2 capsid protein for 30 min at RT followed by secondary antibodies (Jackson ImmunoResearch) for 20 min at RT. The nuclei were stained with DAPI. Image acquisition was carried out using an Axiophot 2 microscope (Zeiss, Oberkochen, Germany). PLS induction in virus-infected cells was always determined using mock-infected cells as control references.

**DENV.** BHK-21 cells (Hamster Kidney Fibroblasts) were cultured in cultured in complete Glasgow MEM medium. Cells were transfected with recombinant DENV2 (strain 16681) RNA using Lipofectamine RNAiMax, according to manufacturer's instructions. Supernatants were collected and spun down to remove debris. DENV2 was propagated in C6/36 *Aedes albopictus* mosquito cells, maintained in Leibovitz L15 medium, by infection with viral supernatants of BHK-21 cells[66] Huh7.5.1[dif] cells were then infected with DENV2 (MOI = 0.5) for 24 h in 1% DMSO-complemented DMEM. Infection was assessed using flow cytometry by intracellular staining of DENV-E protein using mouse anti-DENV-E protein mAb (Millipore)[66]. PLS induction in virus-infected cells was always determined using mock-infected cells as control references.

**Free fatty acid (FFA) and ethanol treatment.** Huh7.5.1[dif] cells were cocultured with 20% LX2 stellate cells for 3 days in DMEM supplemented with 10% heat-decomplemented FBS, gentamycin, and 1% DMSO at 37 °C and 5% $CO_2$. Cells were then incubated with FFA (800 µM oleic acid and 400 µM palmitic acid) for 72 h. For ethanol treatment, Huh7.5.1[dif] cells were incubated in presence or absence of ethanol (20 or 40 mM) for 48 h in a humidified incubator saturated with ethanol vapor. Fresh medium containing ethanol was replenished every day.

PLS induction in FFA- and ethanol-treated cells was always determined using non-treated cells as references.

***Hypoxia treatment and nutrient starvation of Huh7.5.1^dif cells.*** Hypoxia was induced as described[67] with some modifications. Briefly, Huh7.5.1^dif cells were cultured in a humidified incubator supplied with 3% $O_2$, 5% $CO_2$, and 95% $N_2$ for 72 h. Cells cultured in presence of 20% $O_2$ were analyzed as controls. To induce nutrient depletion and starvation, Huh7.5.1^dif cells were incubated in DMEM medium containing low glucose (5.5 mM D-glucose), 1 mM sodium pyruvate, 25 mM HEPES, and 0.1 % heat-decomplemented FBS for 24 h. Cells cultured in medium containing high glucose (25 mM D-glucose) and 10% FBS were used as controls.

***Selection of compounds for the molecular signature-based drug screening.*** Transcriptome-based in silico drug screening was performed using the PLS as a query in the chemogenomics database connectivity map (cmap, www.broadinstitute.org/cmap) and LINCS database (https://clue.io/), as previously described[18,43]. Cmap-predicted compounds with a negative enrichment score and a $p$ value < 0.05, and LINCS-predicted compounds with connectivity score < −90 were further selected with respect to their approval in clinical trials, according to www.clinical trial.gov. Compounds that have been formally licensed for clinical use, compounds currently being evaluated in phases 3–4 clinical trials and compounds with a putative relevance for liver cancer currently being evaluated in phase 2 clinical trials were selected for assessment in the cell-based model. Known carcinogens, antibiotics, and compounds associated with severe liver toxicity according to LiverTox database (http://livertox.nlm.nih.gov) were discarded. To assess the effect of selected compounds on the noninfected and HCV-infected cells in the scRNA-Seq data, gene signatures comprising the targets of each specific compound were obtained by extracting the top 100 upregulated and top 100 downregulated Affymetrix probes as indicated in the LINCS database (BING rowspace). Enrichment of these signatures in association with the HCV viral load in scRNA-Seq data was tested using the preranked GSEA module in GenePattern with Pearson correlation as the rank metric. Probes were collapsed to unique genes by prioritizing landmark over inferred probes and selecting the probe with the highest absolute $z$-score in case of multiple probes. Expression heatmaps were visualized using GENE-E (www.broadinstitute.org/GENE-E).

***Cell-based screen of computationally enriched small molecules.*** Huh7.5.1^dif cells were seeded in 96-well plates and maintained in 1% DMSO-complemented medium. Cells were infected with HCV Jc1 (TCID$_{50}$ = 10$^6$/mL) or cell culture supernatant from mock-electroporated cells for non-infected control wells, as described above. Seven days after infection, cells were incubated with either brefeldin-a (0.1 μM), captopril (1 μM), cediranib (0.5 μM), CI-1040 (1 μM), clomifene citrate (1 μM), dilazep HCl$_2$ (1 μM), dorzolamide (10 μM), erlotinib (0.1 μM), iso-liquiritigenin (10 μM), MK-2206 (1 μM), nizatidine (10 μM), orteronel (40 μM), oxetacaine (10 μM), PD0325901 (10 nM), pimasertib (0.5 μM), pimozide (1 μM), pioglitazone (1 μM), pralidoxime (10 μM), primidone (10 μM), resveratrol (10 μM), rolipram (10 μM), selumetinib (0.5 μM), TG-101348 (0.1 μM), tivozanib (0.5 μM), tolnaftate (10 μM), or triamcinolone (58 nM) for 3 days. The concentration of each test compound was determined based on the lack of cytotoxicity in cell-based model as assed using Presto Blue® reagent, according to manufacturer's recommendations (Life Technologies). After 3 days of treatment, cells were washed with PBS, lysed with 5 μL of iScript™ RT-qPCR Sample Preparation Reagent (Bio-rad), and subjected to nCounter Digital Analyzer system (NanoString).

***Histamine, 8-CPT cAMP, and H89 treatment.*** Huh7.5.1^dif cells were cultured in DMEM containing 1% FBS and 1% DMSO, and incubated with histamine (10 μM) or histamine + nizatidine (10 μM) for 48 h. Fresh medium containing histamine was replenished every 12 h. For 8-CPT cAMP and H89, Huh7.5.1^dif cells cultured in DMEM 1% DMSO and 1% FBS were incubated with 8-CPT cAMP (100 μM), H89 (1 μM), or DMSO as a CTRL for 48 h. Fresh medium containing 8-CPT cAMP was replenished every 12 h.

***In vitro CRISPR/Cas9 gene editing.*** Lentiviruses expressing single guide RNA (sgRNA) were produced in HEK 293T cells by co-transfection with an envelope plasmid (pMD2.G), a packaging plasmid (psPAX2), and a lentiviral vector expressing the sgRNA (pXPR_BRD016) from the Broad Institute. Co-transfection was performed using the CalPhos™ Mammalian Transfection Kit (Clontech Laboratories), according to manufacturer's instructions. Huh7.5.1 stably expressing Cas9 endonuclease (Huh7.5.1-Cas9) were generated by transduction of a lentiviral vector expressing Cas9 (pXPR_BRD111, Broad Institute). For *HRH2* KO, Huh7.5.1-Cas9 cells were then transduced with lentiviruses expressing sgRNA CTRL targeting *GFP* (sgCTRL) or sgRNA targeting *HRH2* (sgHRH2) designed by the Broad Institute. After 48 h, transduced cells were selected under hygromycin treatment (125 μg/mL). HRH2 KO was determined at genetic level using T7 endonuclease assay (Alt-R® Genome Editing Detection Kit, IDT™), according to manufacturer's instructions. For *CREB5* KO, Huh7.5.1-Cas9 were transduced with lentiviruses expressing sgRNA CTRL targeting *GFP* (sgCTRL) or sgRNA targeting

*CREB5* (sgCREB5). After 48 h, transduced cells were selected under hygromycin treatment (125 μg/mL). The KO efficacy was assessed by western blot analysis.

HRH2 KO THP1 cells were engineered by Synthego (Synthego Corporation, Menlo Park, California, USA). Briefly, sgRNA targeting the beginning of HRH2 coding sequence were designed using Synthego CRISPR Design Tool (https://design.synthego.com) and complexed with *Streptococcus pyogenes* Cas9 2NLS nuclease to form ribonucleoproteins before cell transfection. Edited cells were then selected by clonal selection using Sony SH800 Cell Sorter (Sony, Serial number: 0314067).

***Real-time qRT-PCR.*** cDNAs were synthetized by reverse transcription using SuperScript III First-Strand Synthesis SuperMix (Life Technologies). Expression of mouse *Acta2*, *Col1a1*, *Ccl2*, *Cd68*, *Il6*, *Timp1*, and T*gf-β1* was analyzed by quantitative real-time PCR using TaqMan Gene Expression Assays (Thermo Fisher Scientific) on Applied Bioscience 7900HT Fast Real-Time PCR system. Expression of human *HDC*, *IL6*, *IL1β*, *TNFα*, *HRH2*, and *CREB5* was analyzed by quantitative real-time PCR using TaqMan Gene Expression Assays (Thermo Fisher Scientific) on the CFX96 Touch Real-Time PCR Detection System PCR system (Bio-Rad). Expression of human CCL2 and TGF was analyzed using iTaq™ Universal SYBR® Green Supermix (Bio-Rad). The $2^{-\Delta CT}$ method was applied for relative quantification of mRNA with normalization to either 18S or GAPDH mRNA. The list of the primers is provided in Supplementary Table 4.

***Generation of plasmids for HDTVI.*** For different HRH2-targeting sgRNAs, predicted to have high on target efficiency and no off targets, were selected using Chopchop V3 (https://doi.org/10.1093/nar/gkz365). For efficiency testing, sgRNAs targeting *Hrh2* were cloned into the vector pX459, as a dual sgRNA cassette either in combination with a nontargeting control sgRNA (sgCTRL) or in combination with a *Tp53*-targeting sgRNA (sg*Tp53*). In the case of a missing 5′ guanine, one was added as transcription start site for polymerase III-dependent transcription. The cloning procedure was adapted from ref. [68]. pX459 containing a nontargeting sgRNA was used as a control. Plasmids were then transfected into FL83B cells using Lipofectamine 3000 (Invitrogen), according to manufacturer's recommendations. One day post transfection, cells were washed, and from 12 h later on, cells were cultured in the presence of 10 μg/mL puromycin (Carl Roth) for 3 consecutive days. Afterward, DNA was extracted (QiaAmp, Qiagen) and PCR amplicons were submitted to Sanger sequencing (Microsynth Seqlab) for Tracking of Indels by Decomposition (TIDE) analysis. The best performing sgRNA was cloned into pX330 for in vivo delivery via HTVI, either in combination with sgCTRL or sg*Tp53*. pX459 (https://doi.org/10.1038/nprot.2013.143) and pX330 (https://doi.org/10.1126/science.1231143) were generous gifts from Feng Zhang (Addgene plasmid # 62988 and Addgene plasmid # 42230, respectively). The CMV-SB13 and pT3-EF1α-KRASG12D-IRES-EGFP plasmids were kindly provided by Dr. Darjus Tschaharganeh (German Cancer Research Center, Germany). Vectors for hydrodynamic delivery were produced using the QIAGEN Plasmid Maxi Kit (QIAGEN).

***EdU assay and flow cytometry.*** HRH2 KO Huh7.5.1 cell proliferation was assessed using Click-iT® EdU Flow Cytometry Assay Kit (ThermoFischer Scientific), according to manufacturer's instructions. Briefly, *HRH2* KO Huh7.5.1 or CTRL cells were incubated with EdU (5-ethynyl-2′-deoxyuridine) for 3 h at 37 ℃. Then, cells were detached and 200,000 cells per conditions were fixed with PFA 0.5% and permeabilized using 0.2% saponin + 1% FBS before to be incubated with Click-iT® EdU detection cocktail Edu incorporation was analyzed by Flow Cytometry. Data were acquired using Cytoflex B2R2U0 cytometer (Beckman Coulter, BA47394) and CytExpert 2.3 software and then analyzed using FlowJo V10.5.3. Cells incubated without Edu, but exposed to the detection cocktail served as reference. For PHH, HRH2 was stained using rabbit polyclonal HRH2-specific antibody. Cytokeratin 18 (CK18) was used as hepatocyte marker (mouse monoclonal anti-CK18 antibody). Rabbit or mouse control IgG were used as negative controls. Primary antibodies were detected using Alexa FluorTM 647-conjugated rabbit-specific and PE-conjugated mouse-specific secondary antibodies, according to manufacturer's instructions. Data were acquired using Sony SH800 Cell Sorter (Sony, Serial number: 0314067) and SH800 cell sorter software V2.1.5 then analyzed using FlowJo V10.5.3.

***In vitro apoptosis assay.*** Apoptosis level in *HRH2* KO or CTRL Huh7.5.1 cells was assessed by detecting cleaved caspase 3 after $H_2O_2$ treatment (300 μM, 3 h) using CellEvent™ Caspase 3/7 (ThermoFischer Scientific), according to manufacturer's instructions. Immunofluorescence pictures were acquired using Axio Observer Z1 microscope. Quantification of cleaved caspase 3 levels were performed using Celigo Imaging Cytometer. Data were integrated and normalized to total cell number (DAPI staining).

***Histology, immunohistochemistry.*** Animal tissues: formalin-fixed samples were embedded in paraffin, cut in 5 μm thick sections, and stained with hematoxylin and eosin or Picrosirius red staining. The CPA was morphometrically quantified on Sirius red stained sections with image processing software (Image J, NIH).

Additional sections in the DEN injury model were stained with antibody specific for lymphocyte antigen 6 complex locus G6D (Ly6G), F4/80, cluster of differentiation 3 (CD3), cluster of differentiation 4 (CD4), cluster of differentiation 8 (CD8), HRH2, and CREB5. IHC were quantified using Image J IHC ToolBox.

Human tissues: formalin-fixed, paraffin-embedded human tissue samples were provided by the tissue biobank of the University Medicine Mainz in accordance to the ethics committee of the Medical Association of the State of Rhineland Palatinate, Germany (Immunohistochemical analysis of HCC, No. 837.146.17 (10980), April 24th, 2017). Formalin-fixed, paraffin-embedded tissues were cut at a thickness of 1–2 μm to perform immunohistochemistry[69]. Human tissue sections were incubated in an oven at 60 °C for 4 h. After rehydration in graded ethanol, antigen retrieval was performed in 0.1 M Tris-HCl containing 5% urea, pH 9.5 in a steamer at 100 °C for 15 min. Sections were preincubated with 3% $H_2O_2$ for 30 min and blocked with the serum of the species, in which the secondary antibody was raised for 30 min, and then with avidin and biotin for 10 min each. Then, polyclonal rabbit anti-HRH2-serum (GeneTex) was used at a dilution of 1:100 in Tris-buffered saline at pH 6.1 overnight. After subsequent washes in PBS for 20 min, sections were incubated with the secondary antibody for 30 min, washed with PBS, and incubated with the avidin–biotin complex for 10 min. After washing with PBS, sections were incubated with the peroxidase complex for 2–10 min. Sections were mounted with Eukitt (Riedel de Haen, Seelze, Germany). Images were taken with an Olympus BX45 microscope using Plan ×20 objectives with a Gryphax Progress camera (Jenoptik) and Cell Sense software (Micro Optimal, Kirchheim / Teck).

*Single-cell RNA-Seq analyses of patient liver tissues.* Human cells were dissociated from HCC patient liver tissues undergoing surgical resection using gentleMACS™ Octo Dissociator with Heaters and Human tumor dissociation kit (Miltenyi Biotec). After dissociation, cells were filtered using 100 μm MACS SmartStrainers (Miltenyi Biotec) and centrifuged 5 min at 50 g to remove hepatocytes. Supernatants were then centrifuged 10 min at $800 \times g$ and cells were washed with Hank's Balanced Salt Solution (Gibco). Erythrocytes were removed using Buffer EL (Qiagen) and washing in PBS + 1% FBS. Leukocyte CD45+ cells were separated from other cell types by flow cytometry using SH800 cell sorter (Sony). (The gating strategy is described in Supplementary Fig. 15). Living cells were selected using Zombie green staining according to manufacturer's instructions. Data were acquired using the Sony Cell Sorter Software. CD45+ cells were then treated with nizatidine 20 μM or DMSO vehicle control. Two days after treatment, single cells were sorted into 384 well cell capture plates (Single-Cell Discovery, https://www.scdiscoveries.com) using SH800 cell sorter (Sony). Sorted plates were briefly centrifuged at 4 °C, snap-frozen on dry ice and stored at −80 °C until processed. The plates contain mineral oil and droplets of poly-A primers, containing a cell-specific barcode and a unique molecular identifier, enabling to distinguish the well-specific (and cell-specific) mRNA molecules. scRNA-Seq was performed by Single-Cell Discoveries B.V. using SORT-Seq, a modified CEL-Seq2 protocol[38]. Reads were aligned to the human hG19 UCSC reference using Hisat2. R version 3.5.3 with package "RaceID" for clusterization, cluster analysis, and DEG calculation, as described[35]. Dataset were filtered, analyzed, and normalized using RaceID3. As quality control, we removed all the cells with >80% of mitochondrial genes. Transcripts correlating to KCNQ1OT1 with a Pearson's correlation coefficient of >0.4 were removed. RaceID3 was run with the following parameters: mintotal = 1500, minexpr = 2, minnumber = 5, knn = 10. This step filtered out the cells with <1500 transcripts. For normalization, the total transcript counts in each cell were normalized to 1 and multiplied by the minimum total transcript count across all cells that passed the quality control threshold. Data are available at NCBI Gene Expression Omnibus database (https://www.ncbi.nlm.nih.gov/geo/query/acc.cgi?acc=GSE173671).

*Culture of organotypic ex vivo patient liver slices, patient-derived spheroids, and tumorspheroids.* Organotypic liver slices: fresh liver tissue sections (300 μm thick) were made from surgically resected fibrotic livers from liver disease patients, using Krumdieck Tissue Slicer MD6000 (Alabama Research and Development)[8]. The tissues were cultured with nizatidine (50 μM) or DMSO vehicle control for 48 h and harvested for gene expression analysis, as described above.

Patient-derived spheroids: spheroids were generated from liver tissues from patients without liver disease undergoing liver resection for metastasis of colorectal cancer. Tissues were perfused and dissociated as described for PHH. Total cell population, including hepatocytes and NPCs, was used to generated multicellular spheroids in Corning® 96-well Black/Clear Bottom Low Flange Ultra-Low Attachment Microplate (Corning). Spheroids were then treated with FFA and or nizatidine (20 μM) for a total of 3 days before PLS assessment. DMSO was used as negative control.

Patient-derived tumorspheroids: tumorspheroids were generated from patient HCC liver tissues undergoing surgical resection and dissociated using Human Tumor Dissociation Kit, as described above (single cell of patient-derived tissues). Total cell populations, including epithelial (i.e., cancer cells/hepatocytes) and NPCs, were used to generated multicellular tumorspheroids in Corning® 96-well Black/Clear Bottom Low Flange Ultra-Low Attachment Microplate (Corning). After 48 h, HCC-derived spheroids were treated with nizatidine 20 μM, sorafenib 10 μM, or DMSO vehicle control for 4 days. Fresh medium containing DMSO or drugs were added every day. Cell viability was assessed using CellTiter-Glo® (Luminescent Cell Viability Assay), according to manufacturer's instruction.

**Reporting Summary**. Further information on research design is available in the Nature Research Reporting Summary linked to this article.

## Data availability

All data generated in this study are provided in the Supplementary Materials, in Source data file or have been deposited at NCBI Gene Expression Omnibus database. For specific questions, contact the corresponding authors and lead contact, Prof. Thomas F. Baumert (thomas.baumert@unistra.fr).

The genomic dataset generated in this study and presented in Figs. 1, 2, 4g, h, 6, and 8a, as well as Supplementary Figs. 1 and 3 have been deposited at NCBI Gene Expression Omnibus database as a super series under accession code GSE66843 "A cell-based model unravels drivers for hepatocarcinogenesis and targets for clinical chemoprevention". Subseries: GSE66841: gene expression profiles of Huh7.5.1 and HepG2 cells infected with HCV, HBV, HDV, and alcohol, and clinical liver tissues treated with various compounds of coculture with LX2 cells. GSE66842: gene expression profiles of differentiated Huh7.5.1 cells infected with HCV Jc1 clone. GSE81040: high-throughput single-cell RNA-Seq profiling of DMSO-differentiated Huh7.5.1 undergoing or not long-term HCV infection. GSE81801: prognostic liver signature profiles of Jc1-infected Huh7.5.1dif cells treated with various drugs. GSE115473: transcriptome profiles of liver from cirrhotic rat treated with nizatidine. GSE173671: a human liver cell-based system modeling a clinical prognostic liver signature combined with single-cell RNA-Seq for discovery of liver disease therapeutics. GSE169084: bulk RNA sequencing of DMSO-differentiated Huh751 at different time points.

The 186 and 32 PLS gene lists and the full 186 gene signature heatmaps are provided in Supplementary Data 1 and Supplementary Fig. 2. Full immunoblots are provided in Supplementary Figs. 7, 8, 11, and 12. Results of the transcriptome-based in silico drug screening using the PLS as a query in the chemogenomic database connectivity map and LINCS are available within Supplementary information (Tables 1 and 2). The following public databases were used in the study are available on https://www.ncbi.nlm.nih.gov/geo/query: GSE14520: HBV-related liver cancer patient cohort[46]. GSE28619: alcoholic hepatitis patient cohort[47]. GSE49541: NASH patient cohort[48]. GSE54102: HCV-related cirrhosis, US[7]. GSE15654: HCV-related cirrhosis, Italy[6]. GSE31803: NAFLD patient cohort[48]. GSE48452: NASH patient cohort[49]. GSE115469: human liver cellular landscape by single-cell RNA-seq[15]. GSE124395: human liver cell atlas[35]. GSE136103: human liver cirrhosis using single-cell transcriptomics[16]. GSE84346: HCV-infected patient cohort[70]. GSE48452: obesity, NAFLD, and NASH patient cohort[49]. GSE10143: cirrhosis patient cohort[5]. GSE94660: HBV–HCC patient cohort[71]. GSE94399: alcoholic hepatitis patient cohort[72]. Source data are provided with this paper.

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

## Acknowledgements

This work was supported by ARC, Paris and Institut Hospitalo-Universitaire, Strasbourg (TheraHCC1.0 and 2.0 IHUARC IHU201301187 and IHUARC2019 to T.F.B.), the European Union (ERC-AdG-2014-671231-HEPCIR to T.F.B. and Y.H., EU H2020-667273-HEPCAR to T.F.B. and M.H., and INTERREG-IV-Rhin Supérieur-FEDER-Hepato-Regio-Net 2012 to T.F.B. and M.B.Z), ANRS, Paris (2013/108 and ECTZ103701 to T.F.B), NIH (DK099558 to Y. H., and CA233794 to Y.H. and T. F. B; CA140861 to B.C.F.; and CA209940, R21CA209940, and R03AI131066 to N.P. and T.F.B.), Cancer Prevention and Research Institute of Texas (RR180016 to Y.H.), US Department of Defense (W81XWH-16-1-0363 to T.F.B. and Y.H.), the Irma T. Hirschl/Monique Weill-Caulier Trust (Y.H.), and the Foundation of the University of Strasbourg (HEPKIN to T. F. B. and Y. H.) and the Institut Universitaire de France (IUF; T.F.B.). M.H. is supported by an ERC CoG grant (HepatoMetaboPath) and EOS grant and by the Deutsche Forschungsgemeinschaft (DFG, German Research Foundation)—Project-ID 272983813—TRR 179, and Project-ID 314905040 SFB TR209. This work has been published under the framework of the LABEX ANR-10-LABX-0028_HEPSYS and Inserm Plan Cancer and benefits from funding from the state managed by the French National Research Agency as part of the Investments for the future program. We thank Prof. R. Bartenschlager (University of Heidelberg, Germany) for providing plasmids for production of HCVcc Jc1 strains, Dr. F. Chisari (The Scripps Research Institute, La Jolla, CA) for the gift of Huh7.5.1 cells, Dr. A. Patel (MRC Virology Unit, Glasgow, UK) for E2-specific mAb AP33, Dr. J. Taylor (Fox Chase Cancer Center, Philadelphia, CA) for HDV expression plasmids, Dr. F. Habersetzer (Strasbourg University Hospitals) for patient samples for isolation of anti-HDV antibodies and infectious HBV, and Drs. Tscharganeh and Dr. Feng Zhang (Broad Institute of Harvard and MIT) for access to plasmids. We acknowledge the CRB (Centre de Ressources Biologiques-Biological Resource Centre), Strasbourg, France, for the management of patient-derived liver tissues. We thank M. Parnot (Inserm U1110, Strasbourg, France) and S. Prokosch (Heidelberg University, Germany) for excellent technical assistance, Drs. D. Guenot and D. Reita (EA 3430, University of Strasbourg, France) for their assistance with hypoxia treatment experiments, Drs. M. Jovanovic, C. Nusbaum, I. Tirosh (Broad Institute of MIT and Harvard), and Dr. S. L. Friedman (Mount Sinai Hospital) for helpful discussions. We thank Dr. S. Chasserot (Plateforme Imagerie In Vitro—NeuroPôle—Strasbourg, France) for support of confocal microscopy analyses. Finally, we thank Single-Cell Discoveries B.V. team (Utrecht, The Netherlands) for the scRNA-seq service and advice, NanoString Technologies, Inc. (Seattle, WA, USA) for technical advice, and Synthego (Synthego Corporation, Menlo Park, California, USA) for THP1 HRH2 KO cell engineering. Some elements of the figures (Figs. 1a, 3h, 3j, 4a, 5b, 6b, and 8, and Supplementary Figs. 5a, 10a, and 15) have been created with PowerPoint and Servier Medical Art (authorization: https://creativecommons.org/licenses/by/3.0/fr/).

## Author contributions

T.F.B. initiated and coordinated the study. Y.H., B.C.F., and T.F.B. designed experiments and analyzed data. E.C., S.B., N.F., S.L., H.E.S., M.F.-V., T.R., X.S., H.H., N.R., C.P., L.H., S.Z., T.Q., A.V., E.R.V., J.L., M.A.O., F.H.T.D., M.S., R.M., L.W., C.T., S.C.D., V.G.M., D.H., J.H., S.N., A.O., W.-M.S., T.H., R.S., Ar.S., B.S., M.A., L.C., E.S., N.G., A.P.K., M.M., V.D.N., N.B., M.H., and B.C.F. performed experiments, and/or analyzed data. A.K.S, O.R.R, and A.R. contributed to design of scRNA-Seq experiments in cell culture. E.C., F.J., and A.S. analyzed patient single-cell RNA-Seq data. N.P. supervised the computational analyses of scRNA-Seq profiling. T.C. performed computational analyses of scRNA-Seq profiling. X.S., R.S.K., and C.B.B. performed liver tissue experiments and bioinformatic analyses. B.K.S and D.S. performed IHC on human liver tissues. M.F.-V. and T.R. performed HTVI experiments, which were supervised by M.H. K.K., M.E.S., R.T.C., G.G., and P.P. provided reagents. E.F. and P.P. provided patient liver tissues. E.C., S.B., Y.H., B.C.F., M.B.Z., C.S., and T.F.B. wrote the manuscript. B.C.F. and K.K.T. supervised animal experiments (fibrosis/NASH/HCC models). Y.H. supervised the bioinformatic analyses and human liver tissue experiments.

## Competing interests

The University of Strasbourg, Inserm, the IHU Strasbourg and Mount Sinai Hospital have filed a patent application on the clinical gene signature-based human cell culture model and uses thereof with Y.H. and T.F.B. as inventors (WO 2016174130 A1), which has been licensed to Alentis Therapeutics, Basel, Switzerland. The University of Strasbourg and Inserm have filed a patent application on H2 blockers targeting liver macrophages for the prevention and treatment of liver disease and cancer with E.C. and T.F.B. as inventors (PCT/EP2021/072341). A.R. is a co-founder and equity holder of Celsius Therapeutics, an equity holder in Immunitas, and was an SAB member of ThermoFisher Scientific, Syros Pharmaceuticals, Neogene Therapeutics and Asimov until 31 July 2020. Since 1 August 2020, A.R. has been an employee of Genentech. O.R.R has been an employee of Genentech. T.F.B. is founder, advisor, and equity holder in Alentis Therapeutics. Y.H. and C.S. hold equity in Alentis Therapeutics. The other authors declare no competing interests.

## Additional information

[1]Institut National de la Santé et de la Recherche Médicale (Inserm), U1110, Institut de Recherche sur les Maladies Virales et Hépatiques, Strasbourg, France. [2]Université de Strasbourg, Strasbourg, France. [3]Liver Tumor Translational Research Program, Simmons Comprehensive Cancer Center, Division of Digestive and Liver Diseases, Department of Internal Medicine, University of Texas Southwestern Medical Center, Dallas, TX, USA. [4]Division of Gastrointestinal and Oncologic Surgery, Massachusetts General Hospital, Harvard Medical School, Boston, MA, USA. [5]Division of Chronic Inflammation and Cancer, German Cancer Research Center, Heidelberg, Germany. [6]Faculty of Biosciences, Heidelberg University, Heidelberg, Germany. [7]Institut Hospitalo-Universitaire, Pôle Hépato-digestif, Nouvel Hôpital Civil, Strasbourg, France. [8]Department of Biomedicine, University Hospital Basel, University of Basel, Basel, Switzerland. [9]Department of Pathology, Massachusetts General Hospital, Boston, MA, USA. [10]Department of Gastroenterology and Metabolism, Graduate School of Biomedical & Health Sciences, Hiroshima University, Hiroshima, Japan. [11]Department of Genetics and Genomic Sciences, Icahn School of Medicine at Mount Sinai, New York City, NY, USA. [12]Department of Gastroenterological Surgery, Kumamoto University, Kumamoto, Japan. [13]Department of Pharmacological Sciences and Drug Discovery Institute, Icahn School of Medicine at Mount Sinai, New York City, NY, USA. [14]Division of Liver Diseases, Icahn School of Medicine at Mount Sinai, New York City, NY, USA. [15]Department of Neurology, Harvard Medical School, Boston, MA, USA. [16]Broad Institute of Harvard and Massachusetts Institute of Technology, Cambridge, MA, USA. [17]KU Leuven Technology Campus Geel, AdvISe, Geel, Belgium. [18]Liver Center and

Gastrointestinal Division, Massachusetts General Hospital, Harvard Medical School, Boston, MA, USA. [19]Institute of Pathology, University Medicine, Johannes Gutenberg University, Mainz, Germany. [20]Institute for Translational Immunology and Research Center for Immunotherapy (FZI), Johannes Gutenberg University (JGU) Medical Center, Mainz, Germany. [21]Division of Gastroenterology, Beth Israel Deaconess Medical Center, Harvard Medical School, Boston, MA, USA. [22]University of Lille, CNRS, Inserm, CHU Lille, Pasteur Institute of Lille, U1019-UMR 8204-CIIL- Center for Infection and Immunity of Lille, Lille, France. [23]CNRS UPR3572 Immunopathologie et Chimie Thérapeutique, Institut de Biologie Moléculaire et Cellulaire (IBMC), Strasbourg, France. [24]Division of Gastroenterology and Hepatology, Geneva University Hospital, Geneva, Switzerland. [25]Department of Neurology, Icahn School of Medicine at Mount Sinai, New York City, NY, USA. [26]Recanati/Miller Transplantation Institute, Icahn School of Medicine at Mount Sinai, New York City, NY, USA. [27]Massachusetts General Hospital Cancer Center; Harvard Medical School, Cambridge St. CPZN 4216, Boston, MA, USA. [28]Institute for Medical Engineering Science & Department of Chemistry, MIT, Cambridge, MA, USA. [29]Ragon Institute of MGH, MIT and Harvard, Cambridge, MA, USA. [30]Department of Biology, Massachusetts Institute of Technology, Cambridge, MA, USA. [31]Present address: Genentech, 1 DNA Way, South San Francisco, CA, USA. [32]Present address: Cancer Research Center of Lyon (CRCL), UMR Inserm 1052 CNRS 5286 Mixte CLB, Université de Lyon 1 (UCBL1), Lyon, France. [33]Present address: Ferring Pharmaceuticals Inc 4245 Sorrento Valley Blvd, San Diego, CA, USA. ✉email: Bryan.Fuchs@ferring.com; Yujin.Hoshida@UTSouthwestern.edu; thomas.baumert@unistra.fr

