## [Peer Review File · Nature Communications]

REVIEWER COMMENTS

Reviewer #1 (Remarks to the Author): Expert in single-cell RNA-seq

Summary: In this manuscript, the authors present a cell-based chemical screening system (cPLS) for high-throughput identification of chemical compounds for advanced liver diseases and hepatocellular carcinoma (HCC) chemoprevention. They used prognostic liver signature (PLS) calculated from transcriptomic analysis as a readout for liver disease progression and risk of HCC development. After validating the effectiveness of their in vitro system, they identified nizatidine (HRH2 antagonist) as a candidate compound. They showed that nizatidine reverts the poor prognosis of their in vitro system through the HRH2-cAMP-CREB axis, and validate its mechanism of actions and therapeutic effects in animal models, human samples, and ex vivo model.

The present manuscript is potentially important because it provides an in vitro system for high-throughput chemical screening for advanced liver diseases and HCC chemotherapy. Their main claims are well supported by data presented and I'm impressed by amount of work. However, I have a major comment regarding the cellular heterogeneity of the cPLS system, which should be carefully addressed.

Major points:

1. Fig. 1k: Does it mean that Huh7.5.1dif cultured with DMSO is a mixture of hepatocytes and Kupffer cells? This is an important point because it affects the interpretation of screening results with respect to investigate the mechanism of actions of identified compounds. Actually, the authors showed that HRH2, a molecular target of nizatidine identified by the chemical screening based on the cPLS system, is expressed in both hepatocytes and Kupffer cells in vivo. They should examine this possibility more rigorously by analyzing their scRNA-seq shown in Fig. 1e. If the number of cells is not enough for this purpose, they should examine the cellular heterogeneity in Huh7.5.1dif by massively parallel scRNA-seq or flow cytometry. I'm also wondering whether the etiologic agents affect the cellular heterogeneity of the cPLS system. If Huh7.5.1dif is a mixture of multiple cell types, we cannot exclude the possibility that the change of PLS upon etiologic agents might be driven by the change of cell type proportions or cellular cross-talks in the cPLS system.

2. Fig. 2a: For high-throughput chemical screening based on the cPLS system, the authors used 32 genes as a gene set of PLS, but they used the full 186 genes in Figure 1 to validate their system. They should provide a rationale of using this reduced gene set as a readout of PLS.

Minor points:

1. The statistical significance of the correlation between the HCV viral load and the PLS induction in Figure 1e should be evaluated.

2. Figure 4 and 5: The two figures provide the same information. One of them might be better presented in supplementary information.

Reviewer #2 (Remarks to the Author): Expert in HCC genomics

Crouch et al. reported a human liver cell model recapitulating a clinical prognostic signature (PLS)

in liver diseases from chronic liver damages to HCC. Using this model, they attempted to discover novel therapeutic compounds to correct the disease status. They identified nizatidine, an HRH2 blocker, as a potential drug for the treatment of advanced liver diseases and HCC chemoprevention, which was further validated by animal models and patient-derived tumor-spheroids. Single-cell RNAseq uncovered that both liver macrophages and hepatocytes are nizatidine targets.

This study has established a unique cell-based model of human liver diseases for rapid and efficient drug discovery. By combining with the single cell analyses of human liver samples as well as animal models, this model could isolate drug-target cells, which would be beneficial for identifying biomarkers. This study is potentially interesting, however, several critical issues remain.

Major comments

1. Issues on the hepatoma-cell model

Although Huh7.5.1 hepatoma cell system can recapture human liver disease status in vitro, the major issue of this system is whether the presence of genetic and epigenetic alterations of cancer cells may affect the screening results (this may miss better drugs). It seems that an organoid model of non-cancerous human hepatocytes would be superior to their hepatoma model. In Figure 8, the author established 3D cultured human non-cancerous hepatocytes. If the authors stick to use a hepatoma system, they should perform similar screening using a non-cancerous hepatocyte model, and prove that there is no bias on drug selection. The author also should discuss the pros and cons of their hepatoma system regarding this point.

2. Involvement of HRH2 as a target of nizatidine in liver diseases

It is well established that nizatidine blocks HRH2, however, it is possible that an off-target effect of nizatidine may be involved in suppressing the disease status of the human liver. To exclude this possibility, the authors should generate an HRH2 knock out models (both HuH7.5.1. and animal models) to ensure that the effects by nizatidine are completely canceled by the genetic deletion of HRH2.

3. The specificity of the cAMP pathway on the PLS signature

Figure 3 presents that the cAMP pathway plays an important role on the induction of PLS signature. However, the authors identified many drug candidates that altered the PLS signatures by the screening. Do all the candidates target the cAMP pathway? Or some of them target other signaling pathways? If so, why a knocking out of the CREB5 gene showed so robust effect in spite of the presence of alternative signaling to regulate the PLS signature? And is there any synergistic effect by simultaneous administration of both nizatidine and captopril?

4. The contribution of hepatocytes and macrophages to the nizatidine effect in vivo

It is significant that the nizatidine treatment reduced liver fibrosis and HCC development in vivo (Figures 4/5). This also provides an opportunity to determine how the two cell types (hepatocytes and macrophages) contributed to the drug effect. Conditional knockout of the HRH2 gene in hepatocyte or macrophage should be conducted to determine that HRH2 signaling of these cell types truly contributed to the reduction of liver fibrosis and chemoprevention of HCC. This is an important experiment and should not be evaded because no direct evidence that links the HRH2 signaling and HCC chemoprevention was provided in the current study. The association was speculated solely by the gene expression analysis.

Minor comments

1. Page 9, lines 11-14: Fig1K shows that the similarity is highest for the innate lymphoid cell. Mononuclear phagocyte is the second. Please confirm the description.
2. Page 17, lines 8-9: In Figure 4b, there seems that no statistical significance was observed for the Ki-67 staining. If so, "nizatidine decreased expression of the Ki-67 protein" is a misleading description.
3. Page 17, lines 15-16: Please add the P-value for Tgfb-1 and Timp1. Did they show statistical significance?
4. Figure 4 g-h/Page 18 lines 1-2: Why nizatidine treatment decreased HRH2 expression? Does nizatidine block the HRH2 indirectly by reducing the expression? Is this effect previously described in the literature?
5. Page 25, line 6: There is no Fig6j in the PDF. Please check.

Reviewer #3 (Remarks to the Author): Expert in computational drug discovery and screening

The workflow leading to the selection of drug candidates submitted to PLS screening should be better described. 54 candidates were initially selected, presumably from the LINCS screening. How about the top scoring compounds identified by Cmap? Were they taken into consideration? In results of Table S2 (Cmap) and Table S3 (LINCS) there appears to be limited overlap in the type of top-score drugs. Can you comment on this?

The criteria adopted to shorten the list from 54 to 25 candidates are not described with sufficient detail. This operation may have excluded potentially interesting candidates.

CI-1040, one of your top-significant molecules, is a known MEK allosteric inhibitor with potential relevance in hepatocellular carcinoma. A comment on this would be interesting to read

Reviewer #4 (Remarks to the Author): Expert in HCC genomics, organoids, and mouse models

In this manuscript from Crouchet et al., the authors aim at generating a human liver cell-based system that can model a clinical prognostic liver signature (PLS) to predict the progression to HCC. Using this as a model they identify the compound nizatidine as a potential chemotherapeutic agent for HCC. The concept of generating a cellular model with predictive value for the progression towards HCC and that can be used to identify therapeutics is sound, timely and interesting. The identification of nizatidine as a potential therapeutic to prevent fibrosis and subsequent cancer development in vivo is interesting. However, this manuscript suffers from some flaws and wrong data interpretations/claims that impede its publication in the actual form.

My major concerns:

1) the authors use hepatoma cell lines as cellular models. In the first part, they validate the system by inducing differentiation by DMSO treatment. Then they perform genome-wide profiles and claim that the differentiated hepatoma cells have acquired hepatocyte-like phenotype. That experiment has the caveat that the transcriptional profile of the treated cells is compared to previously published transcriptomic data in naïve Huh7.5 cells. While it is ok to compare own's results to publicly available databases, this internal control of time point 0 (w/o DMSO) treatment is necessary and, unfortunately, is missing in Supplementary Fig 1. This reviewer finds this a major point, as the authors cannot know how the data from the published naïve cells differs from their own "naïve

cells”, Just technically, this could impose a bias on the different RNASeq profiles, RNAseq batch effects... Scientifically, it imposes a fundamental problem that the cells could even be treated different between labs. Unfortunately, this data is the basis for the rest of the paper.

2), the different perturbations to show the value of the cell model as prognostic:

In Figure 1 the authors infect the cells with HCV, and from a series of transcriptional profiling experiments conclude that is the viral infection the cause of the induction of the poor-prognosis prognostic liver signature (PLS). Unfortunately, there is not proof for that statement. While it is true that following infection there is a liver signature that indicates poorer prognosis, the control with a mock-infected culture is missing and hence causality cannot be attributed. Similar problem occurs with the other viral infections or the metabolic perturbations, there is no control, so this reviewer assumes that the authors used the publicly available dataset to compare.

3) Similarly, by comparing profiles from healthy and cirrhotic patients the authors conclude that their cell based system captures molecular pathways shared between hepatocytes and liver macrophages and that therefore it can be used to investigate dysregulated pathways in nonhepatocyte cell types. Unfortunately, this again is a wrong comparison as the cirrhotic liver is not only made of the “NPCs/ macrophages” but of other cell types, including remnant epithelial tissue, and hence the similarity between the cell experiments and the tissue data just be because of the latter and not because the in vitro cells recapitulate the signature of macrophages in the cirrhotic tissue.

4) Next the authors use the system above to screen for drugs that revert the phenotype of the PLS. They find 11 hits and focus on the HRH2 inhibitor Nizatidine. From CRISPR Knockdown and inhibitor experiments they conclude that the signalling pathway through HRH2 is driver of PLC. However, to be called driver should be able to initiate the PLS in a naïve cell, and not in a cell already carrying additional mutation, like the hepatoma cell line used Huh7.5.

5) the in vivo results showing that Nizatidine prevents liver cancer in vivo. Is also interesting to see that the authors can validate that data on human samples grown as spheroids. The authors claim that this result is in part due to the effect of the drug directly on the macrophages. However, whether they target the inflammatory tumour niche, in particular the macrophages, directly or it is secondary to targeting the tumour cells is not fully demonstrated by the experiments. Maybe isolating tumour macrophages and treating these with the drug could help elucidate that.

Point-by-point response “A human liver cell-based system modeling a clinical prognostic liver signature combined with single-cell RNA-Seq for discovery of liver disease therapeutics”

NCOMMS-20-28583-T

REVIEWER COMMENTS

Reviewer #1 (Remarks to the Author): Expert in single-cell RNA-seq

Summary: In this manuscript, the authors present a cell-based chemical screening system (cPLS) for high-throughput identification of chemical compounds for advanced liver diseases and hepatocellular carcinoma (HCC) chemoprevention. They used prognostic liver signature (PLS) calculated from transcriptomic analysis as a readout for liver disease progression and risk of HCC development. After validating the effectiveness of their in vitro system, they identified nizatidine (HRH2 antagonist) as a candidate compound. They showed that nizatidine reverts the poor prognosis of their in vitro system through the HRH2-cAMP-CREB axis, and validate its mechanism of actions and therapeutic effects in animal models, human samples, and ex vivo model. The present manuscript is potentially important because it provides an in vitro system for high-throughput chemical screening for advanced liver diseases and HCC chemotherapy. Their main claims are well supported by data presented and I'm impressed by amount of work. However, I have a major comment regarding the cellular heterogeneity of the cPLS system, which should be carefully addressed.

We acknowledge the reviewer for the appreciation of our work and for his/her excellent comments which we addressed by performing additional analyses and modifying the text section.

Major points:

1. Fig. 1k: Does it mean that Huh7.5.1dif cultured with DMSO is a mixture of hepatocytes and Kupffer cells? This is an important point because it affects the interpretation of screening results with respect to investigate the mechanism of actions of identified compounds. Actually, the authors showed that HRH2, a molecular target of nizatidine identified by the chemical screening based on the cPLS system, is expressed in both hepatocytes and Kupffer cells in vivo. They should examine this possibility more rigorously by analyzing their scRNA-seq shown in Fig. 1e. If the number of cells is not enough for this purpose, they should examine the cellular heterogeneity in Huh7.5.1dif by massively parallel scRNA-seq or flow cytometry. I'm also wondering whether the etiologic agents affect the cellular heterogeneity of the cPLS system. If Huh7.5.1dif is a mixture of multiple cell types, we cannot exclude the possibility that the change of PLS upon etiologic agents might be driven by the change of cell type proportions or cellular cross-talks in the cPLS system.

We thank the reviewer for the appreciation of our work and for his/her excellent comments which we carefully addressed both experimentally and by modifying the text sections. In this study, we used both monocellular and multicellular models. Fig. 1k shows a monocellular system based on differentiated Huh7.5.1 only. As requested by the

reviewer, we addressed the heterogeneity in the monocellular Huh7.5.1^{diff} model using our scRNA-Seq data as well as flow cytometry. As an example, we studied gene and protein expression of cytokeratin 18 (CK18), a well-established marker for mature hepatocytes. As predicted by the reviewer and now shown in new panel a-b of Supplementary Fig. 4, scRNA-Seq (t-SNE) and flow cytometry (using an anti-CK18 antibody) analyses confirmed cellular heterogeneity on the single cell level in the monocellular system. In addition to the newly added experimental analyses, we addressed this point by adding the following statement in the text section: “As shown by flow cytometry and scRNA-Seq analyses, the PLS system shows heterogeneity on a cellular level (Supplementary Fig. 4a-b). It is therefore conceivable that the change of PLS upon etiologic agents might be also driven by the change of cell type proportions or cellular cross-talks in the PLS system.” (page 5, lines 23-24; page 6, lines 1-2). Due to space limitations, we moved previous panel 1k into the supplementary data (new Supplementary Fig. 4e-f).

2. Fig. 2a: For high-throughput chemical screening based on the cPLS system, the authors used 32 genes as a gene set of PLS, but they used the full 186 genes in Figure 1 to validate their system. They should provide a rationale of using this reduced gene set as a readout of PLS.

To simplify the protocol, reduce the screening costs and thus increase the clinical translatability of the assay, we used a bioinformatically streamlined version of the PLS comprising 32 genes. This 32-gene version of the PLS is accurate and was validated in multiple patient cohorts in previous studies (King et al., Gut 2015; Nakagawa et al., Cancer Cell 2016). Furthermore, the reduced gene signature has been implemented in an FDA-approved diagnostic assay platform specifically designed for clinical use (Elements assay, NanoString) as Laboratory Developed Test (LDT). The Clinical Laboratory Improvement Amendments (CLIA) lab implementation of the assay is now underway to provide the assay service soon for clinicians worldwide (Geiss et al., Nat. Biotech. 2008; Kojima et al., PLOS ONE 2014; Northcott et al., Acta Neuropathol. 2012; Reis et al., BMC Biotechnol. 2011) (<https://www.utsouthwestern.edu/about-us/administrativeoffices/technology/development/about/>). In the revised manuscript, we better explained and clarified the use of the 32 gene signature as readout in the revised Results section “To simplify the protocol, reduce the screening costs and thus increase the clinical translatability of the assay, we used a bioinformatically streamlined version of the PLS comprising 32 genes bioinformatically defined and validated in multiple patient cohorts in previous studies” (page 11, lines 9-12) as well as the revised Method section on page 42, lines 10-17.

Minor points:

1. The statistical significance of the correlation between the HCV viral load and the PLS induction in Figure 1e should be evaluated.

To address this comment, we further assessed the significance of the correlation between the Pre-ranked GSEA enrichment scores of the PLS signature and the percentage of HCV reads (HCV load). These data are now presented as correlation curves in Fig. 1f and stated in the text on page 5, line 19-23: “Single-cell RNA-Seq (scRNA-Seq) analyses revealed

correlation of HCV viral load and the PLS induction in Huh7.5.1^{diff} cells (Poor-prognosis genes: $R = 0.5099$, $p = 0.0008$; good-prognosis genes: $R = -0.3751$, $p = 0.0171$, Pearson correlation), further supporting the causality of HCV infection for the induction of the poor-prognosis PLS (Fig. 1e-f)."

2. Figure 4 and 5: The two figures provide the same information. One of them might be better presented in supplementary information.

As suggested, the data of the former Figure 4 are now presented in Supplementary Fig. 9.

Reviewer #2 (Remarks to the Author): Expert in HCC genomics

Crouchet et al. reported a human liver cell model recapitulating a clinical prognostic signature (PLS) in liver diseases from chronic liver damages to HCC. Using this model, they attempted to discover novel therapeutic compounds to correct the disease status. They identified nizatidine, an HRH2 blocker, as a potential drug for the treatment of advanced liver diseases and HCC chemoprevention, which was further validated by animal models and patient-derived tumor-spheroids. Single-cell RNAseq uncovered that both liver macrophages and hepatocytes are nizatidine targets. This study has established a unique cell-based model of human liver diseases for rapid and efficient drug discovery. By combining with the single cell analyses of human liver samples as well as animal models, this model could isolate drug-target cells, which would be beneficial for identifying biomarkers. This study is potentially interesting, however, several critical issues remain.

We thank the reviewer for his/her interest in our work and for the excellent and constructive comments. We addressed those by performing additional analyses and improvement of the text as indicated below.

Major comments

1. Issues on the hepatoma-cell model. Although Huh7.5.1 hepatoma cell system can recapture human liver disease status in vitro, the major issue of this system is whether the presence of genetic and epigenetic alterations of cancer cells may affect the screening results (this may miss better drugs). It seems that an organoid model of non-cancerous human hepatocytes would be superior to their hepatoma model. In Figure 8, the author established 3D cultured human non-cancerous hepatocytes. If the authors stick to use a hepatoma system, they should perform similar screening using a non-cancerous hepatocyte model, and prove that there is no bias on drug selection. The author also should discuss the pros and cons of their hepatoma system regarding this point.

We thank the reviewer for the excellent comments which we addressed both experimentally and by modifying the text section. Following the reviewers' suggestion, we established a 3D patient liver spheroid PLS system based on primary human hepatocytes and non-parenchymal cells, which recapitulates the key findings of the hepatoma system such as induction of the PLS and its reversal by the top scoring compound nizatidine (see revised Fig. 3, new panels h-j). Since the 3D liver tissue spheroid model is limited by its

dependency on patient tissue, we used the hepatoma system for screening studies and the 3D patient spheroid system for validation studies (see new panels h-j of revised Fig. 3).

As requested by the reviewer we added a statement in the text addressing the pros and cons of the hepatoma-based model system page 36, lines 17-24, page 37, line 1: *“As any cell culture model, the cPLS platform only partially recapitulates the cell circuits in the diseased liver and represents a simplification of the liver transcriptome. However, the robust induction of the clinical PLS demonstrate that the model is useful to identify compounds reversing the poor prognosis status of the signature (as shown in Fig. 2) or pathways involved in the generation of the signature (Fig. 3). Confirmation of the induction of the PLS in authentic patient tissue-derived spheroids and its reversal by the top-scored compound (Fig. 3h) confirms the validity of the cell line-based model. Whereas the spheroid cPLS model is closer to the patient allowing validation of compounds and mechanisms, the limited availability of patient-derived tissues preclude its application for compound screening.”*

2. Involvement of HRH2 as a target of nizatidine in liver diseases. It is well established that nizatidine blocks HRH2, however, it is possible that an off-target effect of nizatidine may be involved in suppressing the disease status of the human liver. To exclude this possibility, the authors should generate an HRH2 knock out models (both HuH7.5.1. and animal models) to ensure that the effects by nizatidine are completely canceled by the genetic deletion of HRH2.

We thank the reviewer for this excellent suggestion which helped us to decipher the mechanism of action of nizatidine. To address this question, we generated both Huh7.5.1 and animal KO models.

In a large series of functional studies shown in new Figure 5 we observed that HRH2 KO significantly reduces Huh7.5.1 proliferation and increases sensitivity to oxidative stress and apoptosis. Furthermore, liver *Hrh2* KO in a mouse model for hepatocarcinogenesis decreases tumor burden and cell proliferation with significantly improved survival as shown by Prof. Heikenwälders' lab (new Fig. 5). These data demonstrate a functional role of HRH2 in liver disease and hepatocarcinogenesis consolidating that nizatidine indeed mediates its disease-modulating effect through HRH2. Of note, nizatidine was the last of the H2 receptor antagonists developed and is significantly more selective than other drugs of this class. For example, nizatidine is not a potent inhibitor of CYP enzymes like cimetidine, the first H2 receptor antagonist. Nevertheless, our data do not exclude that nizatidine may act through additional targets which remain to be discovered. To address this point, we added the following statement in the revised discussion: *“While our genetic loss-of-function studies demonstrate a functional role for HRH2 in liver disease cell circuits and hepatocarcinogenesis and our functional data in hepatocyte and macrophage cell lines suggest a class-effect of HRH2 inhibitors, we cannot exclude that nizatidine may exert the observed biological effects also via targets other than HRH2.”* (page 38, lines 8-11). In the result section we added: *“Nevertheless, our data do not exclude that additional mechanisms or targets are at play”* (page 23, lines 1-2).”

3. The specificity of the cAMP pathway on the PLS signature

Figure 3 presents that the cAMP pathway plays an important role on the induction of PLS signature. However, the authors identified many drug candidates that altered the PLS signatures by the screening. Do all the candidates target the cAMP pathway? Or some of them target other signaling pathways? If so, why a knocking out of the CREB5 gene showed so robust effect in spite of the presence of alternative signaling to regulate the PLS signature And is there any synergistic effect by simultaneous administration of both nizatidine and captopril?

Each candidate drug identified in our screening targets a specific signaling pathway. As an example, erlotinib targets the Epidermal Growth Factor Receptor (EGFR), captopril targets the Angiotensin Converting Enzyme (ACE), and pioglitazone targets the Peroxisome Proliferator-Activated Receptor Gamma (PPAR γ).

To address the very helpful comment on the signaling pathways, we added the following statements in the results section: *“Interestingly, the top candidate compounds target different and complementary pathways suggesting that several signaling pathways contribute to the induction of the PLS.”* (page 11, lines 17-18).

We focused our efforts to study the functional role of nizatidine on the cAMP-CREB pathway. CREB5 is as key regulator in liver disease biology (Osawa et al., EBioMedicine 2015; Steven et al., Oncotarget 2016) as confirmed by the pronounced PLS induction by CREB5 KO. Furthermore, CREB5 is a transcriptional activator for expression of several of the individual PLS genes such as pro-inflammatory cytokines.

We agree with the reviewer that signaling pathways can also converge and activate the same transcription factors as both, HRH2 and angiotensin receptors, which belongs to GPCR family, may activate CREB (Kk et al., Sci. Rep. 2020; Steven et al., Oncotarget 2016). Indeed, a combination of captopril and nizatidine suggests an additive effect on the PLS induction (Fig. I to the reviewer). We plan to publish the data showing in Fig. I below in a separate manuscript, where we have analyzed in detail the effect of Captopril and the ACE pathways on the PLS, liver fibrosis progressing and hepatocarcinogenesis.

PLS genes	FFA VS Mock	FFA + Niza VS FFA	FFA + Capto VS FFA	FFA + Niza Capto VS FFA
Poor-prognosis	NES: 3.06 FDR: < 0.001	NES: -2.43 FDR: 0.001	NES: -2.39 FDR: 0.001	NES: -2.77 FDR: < 0.001
Good-prognosis	NES: -2.75 FDR: < 0.001	NES: 2.27 FDR: 0.002	NES: 2.25 FDR: 0.002	NES: 2.70 FDR: < 0.001

Figure I. Effect of nizatidine and captopril treatment on the 186 gene PLS in a patient-derived spheroid model. Normalized enrichment score (NES) and associated false discovery rate (FDR).

To address the very helpful comment on CREB5, we added the following statements in the results section: *“The finding that CREB5 KO results in robust reversal of the PLS is most likely due to the fact that CREB5 is a transcription factor activated by several pathways which mediate expression of PLS genes. Furthermore, CREB5 itself is a transcriptional*

activator for expression of several PLS genes. Collectively, our results demonstrate that the histamine/HRH2/cAMP/CREB signaling pathway is a mediator of the cPLS poor prognosis status (Fig. 3g).” page 15, lines 2-7.

4. The contribution of hepatocytes and macrophages to the nizatidine effect in vivo

It is significant that the nizatidine treatment reduced liver fibrosis and HCC development in vivo (Figures 4/5). This also provides an opportunity to determine how the two cell types (hepatocytes and macrophages) contributed to the drug effect. Conditional knockout of the HRH2 gene in hepatocyte or macrophage should be conducted to determine that HRH2 signaling of these cell types truly contributed to the reduction of liver fibrosis and chemoprevention of HCC. This is an important experiment and should not be evaded because no direct evidence that links the HRH2 signaling and HCC chemoprevention was provided in the current study. The association was speculated solely by the gene expression analysis.

We thank the reviewer for this important point. To address this, we deleted HRH2 in Huh7.5.1 and in hepatocytes of a mouse model of hepatocarcinogenesis using Hydrodynamic Tail Vein Injection (HTVI). In a large series of functional studies shown in the new Fig. 5 we observed that HRH2 KO robustly and significantly reduces Huh7.5.1 proliferation and sensitivity to oxidative stress and apoptosis. Furthermore, liver *Hrh2* KO in mice decreases tumor burden, cell proliferation and significantly improved animal survival as shown by Prof. Heikenwälders’ lab (new Fig. 5). These data demonstrate a direct link between HRH2 hepatocyte signaling and hepatocarcinogenesis and suggest that nizatidine mediates its disease modulating effect through HRH2.

We agree that a conditional KO in either hepatocytes or macrophages *in vivo* would be an elegant approach (and we would love to do this !). However, the establishment of a conditional KO (12-15 months) with subsequent investigation of liver disease phenotype (9-12 months) requires a time frame of more than two years and resources which are out of scope of the revision time for this study. Furthermore, the Covid-19 pandemic has resulted in lock-downs of our institutional animal facilities reducing dramatically their capacity.

To address the functional role of HRH2 in macrophages we generated a HRH2 KO THP1 macrophage cell line and assessed their functional phenotype. Interestingly, our new functional studies show that the HRH2 KO resulted a similar modulation of pro-inflammatory cytokines as treatment with nizatidine (new Supplementary Fig. 14). These data suggest that at least a part of the observed immunomodulatory effect of nizatidine in the THP1-derived macrophages is mediated through HRH2 and macrophage HRH2 contributes to the therapeutic effect of nizatidine *in vivo*.

Nevertheless, our data do not exclude nizatidine may act through additional targets which remain to be discovered. To address this point, we added the following statement in the revised Discussion section: “While our genetic loss-of-function studies demonstrate a functional role for HRH2 in liver disease cell circuits and hepatocarcinogenesis and our functional data in hepatocyte and macrophage cell lines suggest a class-effect of HRH2

inhibitors, we cannot exclude that nizatidine may exert the observed biological effects also via targets other than HRH2.” (page 38, lines 9-11). In the Results section we added: *“Nevertheless, our data do not exclude that additional mechanisms or targets are at play”* (page 23, lines 1-2). Finally, we toned-down the abstract conclusion by adding the term *“potential”*. The revised statement on macrophages reads: *“...liver macrophages as potential nizatidine targets.”*

Minor comments

1. Page 9, lines 11-14: *Fig1K shows that the similarity is highest for the innate lymphoid cell. Mononuclear phagocyte is the second. Please confirm the description.*

We modified the text according to the reviewer’s comment: “We observed that the PLS expressing cells share dysregulated pathways with epithelial cells mainly consisting of hepatocytes and with some NPCs including immune cells” (page 10, lines 14-16). Due to space issues, Fig. 1k has been moved to the Supplementary data section (now Supplementary Fig. 4e-f).

2. Page 17, lines 8-9: *In Figure 4b, there seems that no statistical significance was observed for the Ki-67 staining. If so, "nizatidine decreased expression of the Ki-67 protein" is a misleading description.*

Indeed, while Ki-67 was highly and significantly decreased by nizatidine treatment in our NASH/HCC mouse model (Supplementary Figure 9), the observed nizatidine-induced decrease of Ki-67 in DEN-treated animals did not reach significance. We therefore modified the statement on page 18, lines 8-9 to: “Moreover, nizatidine tend to decrease expression of the Ki-67 protein in the liver, a marker associated with proliferation of mammalian cells”.

3. Page 17, lines 15-16: *Please add the P-value for Tgfb-1 and Timp1. Did they show statistical significance?*

Individual values are now presented in the graph to appreciate the distribution of the data. The decrease in Tgfb1 and Timp1 expression did not reach significance.

4. Figure 4 g-h/Page 18 lines 1-2: *Why nizatidine treatment decreased HRH2 expression? Does nizatidine block the HRH2 indirectly by reducing the expression? Is this effect previously described in the literature?*

To address the direct relationship of nizatidine treatment and HRHR2 expression, we treated liver and macrophage cell lines with nizatidine and quantified HRH2 expression after treatment. We observed that nizatidine had no direct effect on HRH2 expression. We therefore assume that the observed down regulation of HRH2 in liver of animals, is due to the improvement of liver disease which results in downregulation of HRH2 in an indirect manner. The results are presented in new Supplementary Figure 8. To address

this valuable comment, we added the following statement in the text: *“To address a direct relationship of nizatidine treatment and HRH2 expression, we treated cell lines with nizatidine and quantified HRH2 expression after treatment. As nizatidine had no effect on HRH2 expression in cell lines or 3D spheroids (Fig. 3i, Supplementary Fig. 8), we assume that the observed down regulation of HRH2 in nizatidine-treated animals is most likely not due to a direct effect of nizatidine on HRH2 expression, but rather due to the improvement of liver disease.”* (page 19, lines 8-13).

5. Page 25, line 6: There is no Fig6j in the PDF. Please check.

As requested, this has been corrected.

Reviewer #3 (Remarks to the Author): Expert in computational drug discovery and screening

The workflow leading to the selection of drug candidates submitted to PLS screening should be better described. 54 candidates were initially selected, presumably from the LINCS screening. How about the top scoring compounds identified by Cmap? Were they taken into consideration?

In results of Table S2 (Cmap) and Table S3 (LINCS) there appears to be limited overlap in the type of top-score drugs. Can you comment on this?

The criteria adopted to shorten the list from 54 to 25 candidates are not described with sufficient detail. This operation may have excluded potentially interesting candidates.

We thank the reviewer for his/her constructive comments and suggestions. The screening was performed using the PLS genes as a query in both the Cmap and the LINCS databases. The LINCS database is more recent and more updated with more cell lines and drugs compared to the Cmap database, explaining the identification of other candidate drugs (Keenan et al., Cell Syst. 2018; Lamb et al., Science 2006). The 25 candidates were then selected from both the Cmap and the LINCS lists based on several criteria: (i) formally licensed for clinical use, (ii) compounds in phase 3-4 clinical trials, (iii) putative relevance for liver cancer and in phase 2 clinical trials, (iv) absent evident toxicity according to LiverTox database (<http://livertox.nlm.nih.gov>). We clarified this in the Material and Method section and in the revised text (page 46, lines 22-23, page 47, lines 1-18, page 11, lines 1-6). By providing the full list of compounds and their ranking (Supplementary Tables 2-3) the readers of our study are able to test additional compounds of the list in the future.

CI-1040, one of your top-significant molecules, is a known MEK allosteric inhibitor with potential relevance in hepatocellular carcinoma. A comment on this would be interesting to read

To address this very valuable suggestion, we added the following comment: *“Interestingly, our screen uncovered the MEK inhibitors CI-1040, PD0325901 and Selumetinib as candidate compounds. MEK is a protein kinase involved in the Ras/Raf/MEK/ERK signaling pathway, which has been shown to play a functional role in HCC biology through regulation*

of apoptosis, cell cycle, cell migration, differentiation, and proliferation and MEK inhibitors are currently in clinical investigation for HCC treatment” (page 12, lines 7-11).

Reviewer #4 (Remarks to the Author): Expert in HCC genomics, organoids, and mouse models

In this manuscript from Crouchet et al., the authors aim at generating a human liver cell-based system that can model a clinical prognostic liver signature (PLS) to predict the progression to HCC. Using this as a model they identify the compound nizatidine as a potential chemotherapeutic agent for HCC. The concept of generating a cellular model with predictive value for the progression towards HCC and that can be used to identify therapeutics is sound, timely and interesting. The identification of nizatidine as a potential therapeutic to prevent fibrosis and subsequent cancer development in vivo is interesting. However, this manuscript suffers from some flaws and wrong data interpretations/claims that impede its publication in the actual form.

We acknowledge the reviewer for the appreciation of our study and addressed the reviewers’ comments by a large series of additional experiments and change of the text section.

My major concerns:

1) the authors use hepatoma cell lines as cellular models. In the first part, they validate the system by inducing differentiation by DMSO treatment. Then they perform genome-wide profiles and claim that the differentiated hepatoma cells have acquired hepatocyte-like phenotype. That experiment has the caveat that the transcriptional profile of the treated cells is compared to previously published transcriptomic data in naïve Huh7.5 cells. While it is ok to compare own’s results to publicly available databases, this internal control of time point 0 (w/o DMSO) treatment is necessary and, unfortunately, is missing in Supplementary Fig 1. This reviewer finds this a major point, as the authors cannot know how the data from the published naïve cells differs from their own “naïve cells”, Just technically, this could impose a bias on the different RNASeq profiles, RNAseq batch effects... Scientifically, it imposes a fundamental problem that the cells could even be treated different between labs. Unfortunately, this data is the basis for the rest of the paper.

To address this valuable comment, we repeated the full DMSO-differentiation kinetic experiment by including naïve Huh7.5.1 cells (time point 0 w/o DMSO) as control. The repeated data using the internal control confirmed the previous results of a shift in the global transcriptome of Huh7.5.1^{diff} cells during DMSO-differentiation to a “hepatocyte-like profile”. This new data set is now presented in Supplementary Fig. 1 of the revised manuscript.

2), the different perturbations to show the value of the cell model as prognostic:

In Figure 1 the authors infect the cells with HCV, and from a series of transcriptional profiling experiments conclude that is the viral infection the cause of the induction of the poor-prognosis prognostic liver signature (PLS). Unfortunately, there is not proof for that statement. While it

is true that following infection there is a liver signature that indicates poorer prognosis, the control with a mock-infected culture is missing and hence causality cannot be attributed. Similar problem occurs with the other viral infections or the metabolic perturbations, there is no control, so this reviewer assumes that the authors used the publicly available dataset to compare.

We thank the reviewer for this important point which allows to clarify that the gene expression data in each experiment are based on the perturbation of gene expression relative to controls. To illustrate the details of the controls, we now provide the original gene expression data as normalized gene expression of the mock-infected or untreated cells (Supplementary Fig. 2).

The PLS calculation method is based on Gene Set Enrichment Analysis (GSEA). By definition, GSEA compares set of genes and shows statistically significant differences between two biological states, i.e., treated versus non-treated cells (Subramanian et al., PNAS 2005). Thus, the data of the “mock” or “control” sample/state are included in the GSEA calculation which shows the difference between the diseased/perturbed state and the control. This is a state-of-the-art representation of gene expression in patient tissues, animal tissues and cell lines (Subramanian et al., PNAS 2005), Hoshida et al., New Engl. J. Med 2008, Gastroenterology 2013; Lupberger et al., Gastroenterology 2019; Nakagawa et al., Cancer Cell 2016, Juehling et al. Gut 2021).

To further clarify that data shown in heatmaps represent the difference of the disease state versus the non-disease/biological control we added in each figure the respective reference control in the legend describing the heatmap (e. g. “HCV-infected versus Mock control”).

Finally, we provide an additional explanation in the revised text on page 5, lines 11-14: “The global induction or suppression of the PLS signature was then determined by Gene Set Enrichment Analysis (GSEA) using mock-infected cells as reference and results were presented as simplified heatmaps (Fig. 1d)”.

3) Similarly, by comparing profiles from healthy and cirrhotic patients the authors conclude that their cell based system captures molecular pathways shared between hepatocytes and liver macrophages and that therefore it can be used to investigate dysregulated pathways in nonhepatocyte cell types. Unfortunately, this again is a wrong comparison as the cirrhotic liver is not only made of the “NPCs/ macrophages” but of other cell types, including remnant epithelial tissue, and hence the similarity between the cell experiments and the tissue data just be because of the latter and not because the in vitro cells recapitulate the signature of macrophages in the cirrhotic tissue.

We thank the reviewer for this important point allowing to improve the manuscript by additional analyses and modifying the text section.

We agree with the reviewer that the cPLS system models cell circuits of the entire liver tissue of patients. To better illustrate this concept, we used public data bases of human liver tissues (MacParland et al., Nature Com. 2018) to determine in which cell types the expression of PLS genes is dysregulated. As shown in Fig. 1j, the PLS poor-

prognosis genes are most prominently expressed in NPCs, whereas the expression of the PLS good-prognosis genes is higher in hepatocytes. Taken together, these data demonstrate that the cPLS system captures dysregulation of molecular pathways of epithelial cells such as hepatocytes as well as certain pathways shared between hepatocytes and NPCs.

The modeling of certain NPCs' cell circuits in the cPLS system was then confirmed by a comparative transcriptomic analysis with a state-of-the-art scRNA-Seq data set comparing the main cell populations described within cirrhotic livers (Ramachandran et al., Nature 2019) and the PLS expressing cells (now shown in Supplementary Fig. 4e-f). Of note, the scRNA-Seq analyses previously shown in Fig. 1k and now presented in Supplementary Fig. 4e-F was favorably reviewed by reviewer #1, a scRNA-Seq expert.

To completely address the comments of the reviewer, we now present the deconvolution of the PLS across all cell types in liver disease in new figure panel 1j, more accurately describe the similarity of the cPLS system to epithelial cells (now Supplementary Fig. 4e-f) and modified our conclusion regarding the similarity of cell phenotypes. The new text section now reads: *“We observed that the PLS expressing cells share dysregulated pathways with epithelial cells mainly consisting of hepatocytes and with some NPCs including immune cells (Supplementary Fig. 4e-f). Together, these data demonstrate that the cPLS system captures molecular pathways of epithelial cells such as hepatocytes as well as certain pathways shared between hepatocytes and NPCs.”* (page 10, lines 14-19). Finally, to completely address the reviewers' comment and tone-down our conclusions on the similarity of the cPLS system and macrophages, we deleted the respective paragraph in the Results section.

4) Next the authors use the system above to screen for drugs that revert the phenotype of the PLS. They find 11 hits and focus on the HRH2 inhibitor Nizatidine. From CRISPR Knockdown and inhibitor experiments they conclude that the signalling pathway through HRH2 is driver of PLC. However, to be called driver should be able to initiate the PLS in a naïve cell, and not in a cell already carrying additional mutation, like the hepatoma cell line used Huh7.5.

To address this question, we performed additional experiments using naïve patient-derived spheroids containing primary hepatocytes and NPCs. Our data demonstrate that the poor-prognosis status PLS is robustly induced by histamine treatment and reversed by the HRH2 inhibitor nizatidine (Figure 3h-j). These data clearly show that the histamine – HRH2 pathway is sufficient to induce the poor-prognosis-status of the PLS in naïve cells. Nevertheless, to interpret our results as careful as possible we tone-down our conclusion by terming the histamine-HRH2 pathway not a “driver” but a “mediator” of the PLS poor-prognosis status (page 15, line 7).

5) the in vivo results showing that Nizatidine prevents liver cancer in vivo. Is also interesting to see that the authors can validate that data on human samples grown as spheroids. The authors claim that this result is in part due to the effect of the drug directly on the macrophages. However, whether they target the inflammatory tumour niche, in particular the macrophages,

directly or it is secondary to targeting the tumour cells is not fully demonstrated by the experiments. Maybe isolating tumour macrophages and treating these with the drug could help elucidate that.

To address reviewer's comment, we isolated tumor-associated macrophages (TAMs) from patient tumor tissues and treated them with nizatidine. We observed that nizatidine decreases IL6 expression in TAMs, as observed in THP1- and patient-derived macrophages, confirming that macrophages are nizatidine targets (Supplementary figure 14).

REFERENCES

- Geiss, G.K., Bumgarner, R.E., Birditt, B., Dahl, T., Dowidar, N., Dunaway, D.L., Fell, H.P., Ferree, S., George, R.D., Grogan, T., et al. (2008). Direct multiplexed measurement of gene expression with color-coded probe pairs. *Nat. Biotechnol.* *26*, 317–325.
- Hoshida, Y., Villanueva, A., Kobayashi, M., Peix, J., Chiang, D.Y., Camargo, A., Gupta, S., Moore, J., Wrobel, M.J., Lerner, J., et al. (2008). Gene Expression in Fixed Tissues and Outcome in Hepatocellular Carcinoma. *N. Engl. J. Med.* *359*, 1995–2004.
- Hoshida, Y., Villanueva, A., Sangiovanni, A., Sole, M., Hur, C., Andersson, K.L., Chung, R.T., Gould, J., Kojima, K., Gupta, S., et al. (2013). Prognostic gene expression signature for patients with hepatitis C-related early-stage cirrhosis. *Gastroenterology* *144*, 1024–1030.
- Keenan, A.B., Jenkins, S.L., Jagodnik, K.M., Koplev, S., He, E., Torre, D., Wang, Z., Dohlman, A.B., Silverstein, M.C., Lachmann, A., et al. (2018). The Library of Integrated Network-based Cellular Signatures (LINCS) NIH Program: System-level Cataloging of Human Cells Response to Perturbations. *Cell Syst.* *6*, 13–24.
- King, L.Y., Canasto-Chibuque, C., Johnson, K.B., Yip, S., Chen, X., Kojima, K., Deshmukh, M., Venkatesh, A., Tan, P.S., Sun, X., et al. (2015). A genomic and clinical prognostic index for hepatitis C-related early-stage cirrhosis that predicts clinical deterioration. *Gut* *64*, 1296.
- Kk, A., P, K., R, G., R, S., H, Z., K, P., C, N., S, L., and Kn, P. (2020). Angiotensin II represses Npr1 expression and receptor function by recruitment of transcription factors CREB and HSF-4a and activation of HDACs. *Sci. Rep.* *10*.
- Kojima, K., April, C., Canasto-Chibuque, C., Chen, X., Deshmukh, M., Venkatesh, A., Tan, P.S., Kobayashi, M., Kumada, H., Fan, J.-B., et al. (2014). Transcriptome Profiling of Archived Sectioned Formalin-Fixed Paraffin-Embedded (AS-FFPE) Tissue for Disease Classification. *PLOS ONE* *9*, e86961.
- Lamb, J., Crawford, E.D., Peck, D., Modell, J.W., Blat, I.C., Wrobel, M.J., Lerner, J., Brunet, J.-P., Subramanian, A., Ross, K.N., et al. (2006). The Connectivity Map: using gene-expression signatures to connect small molecules, genes, and disease. *Science* *313*, 1929–1935.
- Lupberger, J., Croonenborghs, T., Suarez, A.A.R., Renne, N.V., Jühling, F., Oudot, M.A., Virzì, A., Bandiera, S., Jamey, C., Meszaros, G., et al. (2019). Combined Analysis of Metabolomes, Proteomes, and Transcriptomes of Hepatitis C Virus–Infected Cells and Liver to Identify Pathways Associated With Disease Development. *Gastroenterology* *157*, 537–551.e9.

- MacParland, S.A., Liu, J.C., Ma, X.-Z., Innes, B.T., Bartczak, A.M., Gage, B.K., Manuel, J., Khuu, N., Echeverri, J., Linares, I., et al. (2018). Single cell RNA sequencing of human liver reveals distinct intrahepatic macrophage populations. *Nat. Commun.* *9*, 4383.
- Nakagawa, S., Wei, L., Song, W.M., Higashi, T., Ghoshal, S., Kim, R.S., Bian, C.B., Yamada, S., Sun, X., Venkatesh, A., et al. (2016). Molecular Liver Cancer Prevention in Cirrhosis by Organ Transcriptome Analysis and Lysophosphatidic Acid Pathway Inhibition. *Cancer Cell* *30*, 879–890.
- Northcott, P.A., Shih, D.J.H., Remke, M., Cho, Y.-J., Kool, M., Hawkins, C., Eberhart, C.G., Dubuc, A., Guettouche, T., Cardentey, Y., et al. (2012). Rapid, reliable, and reproducible molecular sub-grouping of clinical medulloblastoma samples. *Acta Neuropathol. (Berl.)* *123*, 615–626.
- Osawa, Y., Oboki, K., Imamura, J., Kojika, E., Hayashi, Y., Hishima, T., Saibara, T., Shibasaki, F., Kohara, M., and Kimura, K. (2015). Inhibition of Cyclic Adenosine Monophosphate (cAMP)-response Element-binding Protein (CREB)-binding Protein (CBP)/ β -Catenin Reduces Liver Fibrosis in Mice. *EBioMedicine* *2*, 1751–1758.
- Reis, P.P., Waldron, L., Goswami, R.S., Xu, W., Xuan, Y., Perez-Ordonez, B., Gullane, P., Irish, J., Jurisica, I., and Kamel-Reid, S. (2011). mRNA transcript quantification in archival samples using multiplexed, color-coded probes. *BMC Biotechnol.* *11*, 46.
- Steven, A., Seliger, B., Steven, A., and Seliger, B. (2016). Control of CREB expression in tumors: from molecular mechanisms and signal transduction pathways to therapeutic target. *Oncotarget* *7*, 35454–35465.
- Subramanian, A., Tamayo, P., Mootha, V.K., Mukherjee, S., Ebert, B.L., Gillette, M.A., Paulovich, A., Pomeroy, S.L., Golub, T.R., Lander, E.S., et al. (2005). Gene set enrichment analysis: A knowledge-based approach for interpreting genome-wide expression profiles. *Proc. Natl. Acad. Sci.* *102*, 15545–15550.

REVIEWERS' COMMENTS

Reviewer #1 (Remarks to the Author):

The revised manuscript addressed most of my concerns except the following point:

1. The authors showed that there exist at least two subpopulations (KRT18^{hi} and KRT18^{low}) within Huh7.5.1diff hepatocytes (Fig. S4a and S4b). Do the two subpopulations show the differential PLS global status or differential expression of PLS prognosis genes? In Fig. 1j, the authors demonstrated that good-prognosis genes are upregulated in hepatocytes but poor-prognosis genes are upregulated in non-parenchymal cells of human liver. How does this finding explain the increased expression of poor-prognosis genes upon etiological agent treatment in the cPLS model not containing non-parenchymal cells? Does the KRT18^{low} subpopulation show the characteristics of dysregulated non-parenchymal cells? Does the etiological agent treatment change the cellular heterogeneity of the cPLS model? All these questions should be carefully addressed by generating large-scale scRNA-seq data (control vs etiological agent treatment). The number of cells in the current scRNA-seq data is not enough for addressing these questions.

Reviewer #2 (Remarks to the Author):

The authors appropriately responded to the comments raised by reviewer 2 except one (Major point 4). Additional experiments strongly support and validate the authors' claims.

One minor point:

In response to Major point 1, the revised manuscript says " Confirmation of the induction of the PLS in authentic patient tissue-derived spheroids and its reversal by the top-scored compound (Fig. 3h) confirms the validity of the cell line-based model..." (Page 36). It would be better to re-write the description "the confirmation confirms the model".

Reviewer #3 (Remarks to the Author):

The manuscript can be published in the present form

Reviewer #4 (Remarks to the Author):

the authors have addressed all my comments
Please note that the figures are all blurry.

Point-by-point response “A human liver cell-based system modeling a clinical prognostic liver signature combined with single-cell RNA-Seq for discovery of liver disease therapeutics”

Reviewer 1:

1. The authors showed that there exist at least two subpopulations (KRT18^{hi} and KRT18^{low}) within Huh7.5.1^{diff} hepatocytes (Fig. S4a and S4b). Do the two subpopulations show the differential PLS global status or differential expression of PLS prognosis genes? In Fig. 1j, the authors demonstrated that good-prognosis genes are upregulated in hepatocytes, but poor-prognosis genes are upregulated in non-parenchymal cells of human liver. How does this finding explain the increased expression of poor-prognosis genes upon etiological agent treatment in the cPLS model not containing non-parenchymal cells? Does the KRT18^{low} subpopulation show the characteristics of dysregulated non-parenchymal cells? Does the etiological agent treatment change the cellular heterogeneity of the cPLS model? All these questions should be carefully addressed by generating large-scale scRNA-seq data (control vs etiological agent treatment). The number of cells in the current scRNA-seq data is not enough for addressing these questions.

We thank the reviewer for these helpful suggestions. As requested by the editor, we carefully addressed this question by additional scRNA analyses of the detailed existing scRNA data set based on the high resolution SMARTSeq2 pipeline of the Broad Institute. These new data are now presented in the modified results and discussion section and new panels f-g of Supplementary Fig. 5 as requested by the editor.

To further investigate cellular heterogeneity, we assessed the diversity across individual HCV-infected Huh7.5.1^{dif} cells according to *KRT18* expression levels as suggested by the reviewer. While the sample size was limited (21 single cells), the diverse *KRT18* expression across the cells enabled a robust comparative analysis. First, there was no biased expression of the poor- and good-prognosis-associated genes according to the *KRT18* expression levels, suggesting that there is no differentiation of the HCV-infected Huh7.5.1^{dif} cells to form either parenchymal- or non-parenchymal-cell-like subpopulation with differential expression of the poor- or good-prognosis-associated PLS member genes (Supplementary figure 5f). Second, when we assessed similarity of global transcriptome between each of the HCV-infected Huh7.5.1^{dif} cells and the major liver cell types in human cirrhotic livers. Interestingly, all Huh7.5.1^{dif} cells showed similar magnitude of resemblance to both parenchymal and non-parenchymal cell types irrespective of the *KRT18* expression levels and the HCV viral load (Supplementary figure 5g). These data support that HCV infection, as an example of etiologic agent to induce PLS, does not result in cellular heterogeneity with regard to induction of the parenchymal- or non-parenchymal-cell-associated PLS genes, i.e., the cPLS system at least partially models transcriptional program of both parenchymal and non-parenchymal liver cell types at comparable level across the Huh7.5.1^{dif} cells. These new findings are now shown in Supplementary Fig. 5f and the revised results section (page 9, lines 7-23).

Nevertheless, the resemblance of the Huh7.5.1^{dif} cells to both parenchymal and non-parenchymal cells was not perfect similarity, and this is noted as a limitation of the cPLS model in revised discussion as follows: "As any cell culture model, the cPLS platform only partially recapitulates the cell circuits of the major liver cell types in the diseased liver in a simplified manner, and therefore the platform should be used for early compound screening followed by validation in multi-cell-type experimental systems such as patient-derived spheroids as well as rodent models of advanced liver disease as demonstrated in this study" (page 22, lines 5-10).

Reviewer #2 (Remarks to the Author):

The authors appropriately responded to the comments raised by reviewer 2 except one (Major point 4). Additional experiments strongly support and validate the authors' claims.

One minor point:

In response to Major point 1, the revised manuscript says " Confirmation of the induction of the PLS in authentic patient tissue-derived spheroids and its reversal by the top-scored compound (Fig. 3h) confirms the validity of the cell line-based model..." (Page 36). It would be better to re-write the description "the confirmation confirms the model".

The sentence has been changed: Induction of the PLS in authentic patient tissue-derived spheroids and its reversal by the top-scored compound (Fig. 3h) confirms the validity of the cell line-based model (page 22, lines 13-14).

Reviewer #3 (Remarks to the Author): The manuscript can be published in the present form.

Reviewer #4 (Remarks to the Author): the authors have addressed all my comments. Please note that the figures are all blurry.

The figures are now provided at high resolution.